



# Measurements of biogenic volatile organic compounds at a grazed savannah-grassland-agriculture landscape in South Africa

**Kerneels Jaars[1], Pieter G. van Zyl[1], Johan P. Beukes[1], Heidi Hellén[2], Ville Vakkari[2], Micky Josipovic[1], Andrew D. Venter[1], Matti Räsänen[3], Leandra Knoetze[1], Dirk P. Cilliers[1], Stefan J. Siebert[1], Markku Kulmala[3], Janne Rinne[4], Alex Guenther[5], Lauri Laakso[1,2] and Hannele Hakola[2]**

[1]Unit for Environmental Sciences and Management, North-West University, Potchefstroom, South Africa

[2]Finnish Meteorological Institute, PL 503, 00101 Helsinki, Finland

[3]Department of Physics, University of Helsinki, Finland

[4]Department of Physical Geography and Ecosystem Science Lund University Sölvegatan 12 S-223 62 Lund, Sweden

[5]Department of Earth System Science, University of California, Irvine, USA

Correspondence to: P.G. van Zyl (pieter.vanzyl@nwu.ac.za)

## Abstract

Biogenic volatile organic compounds (BVOCs) are important role players in the chemistry of the troposphere, especially in the formation of tropospheric ozone ($O_3$) and secondary organic aerosols (SOA). Ecosystems produce and emit a large number of BVOCs. It is estimated on a global scale that approximately 90 % of annual VOC emissions are BVOCs. In this study, measurements of BVOCs were conducted at the Welgegund measurement station (South Africa), which is considered to be a regionally representative background site situated in savannah grassland. Very few BVOC measurements exist for grassland savannah and results presented in this study are the most extensive for this type of landscape. Samples were collected twice a week for two hours during daytime and two hours during night-time through two long-term sampling campaigns from February 2011 to February 2012 and from December 2013 to



February 2015. Individual BVOCs were identified and quantified using a thermal desorption
instrument, connected to a gas chromatograph and a mass selective detector. The annual
median concentrations of isoprene, 2-methyl-3-butene-2-ol (MBO), monoterpenes and
sesquiterpenes (SQT) during the first campaign were 14, 7, 120 and 8 pptv, respectively, and
14, 4, 83 and 4 pptv, respectively, during the second campaign. The sum of the concentrations
of the monoterpenes were at least an order of magnitude higher than the concentrations of other
BVOC species during both sampling campaigns, with α-pinene being the most abundant
species. The highest BVOC concentrations were observed during the wet season and elevated
soil moisture was associated with increased BVOC concentrations. However, comparisons
with measurements conducted at other landscapes in southern Africa and the rest of the world
that have more woody vegetation indicated that BVOC concentrations were, in general,
significantly lower. Furthermore, BVOC concentrations were an order of magnitude lower
compared to total aromatic concentrations measured at Welgegund. An analysis of
concentrations by wind direction indicated that isoprene concentrations were higher from the
western direction, while wind direction did not indicate any significant differences in the
concentrations of the other BVOC species. Statistical analysis indicated that soil moisture had
the most significant impact on atmospheric levels of MBO, monoterpenes and SQT
concentrations, while temperature had the greatest influence on isoprene levels. The combined
$O_3$ formation potentials of all the BVOCs measured calculated with MIR coefficients during
the first and second campaign were 1162 and 1022 pptv, respectively. α-Pinene and limonene
had the highest reaction rates with $O_3$, while isoprene exhibited relatively small contributions
to $O_3$ depletion. Limonene, α-pinene and terpinolene had the largest contributions to the OH-
reactivity of BVOCs measured at Welgegund for all of the months during both sampling
campaigns.

## 1  Introduction

Ecosystems produce and emit a large number of biogenic volatile organic compounds (BVOCs)
that are involved in plant growth and reproduction. These species also act as defensive
compounds, e.g. enhancing tolerance to heat and oxidative stress (Sharkey and Yeh, 2001;
Loreto and Schnitzler, 2010), preventing the colonisation of pathogens after wounding, and
deterring insects or recruiting natural enemies of herbivores (Holopainen and Gershenzon,
2010). The BVOC production rate in an ecosystem depends on several physical (e.g.
temperature, precipitation, moisture, solar radiation and $CO_2$ concentration) and biological



parameters (e.g. plant species and the associated emission capacity, phenology, biotic and
abiotic stresses, attraction of pollinators) (Blande et al., 2014; Fuentes et al., 2000; Kesselmeier
and Staudt, 1999; Sharkey and Yeh, 2001), with typically 0.2 to 10 % of the carbon uptake
during photosynthesis being converted to BVOCs (Kesselmeier et al., 2002).  It is estimated
that, on a global scale, approximately 90 % of annual VOC emissions are by vegetation
(~1000 Tg C year$^{-1}$) (Guenther et al., 2012).
BVOCs can contribute significantly to the carbon balance in certain ecosystems (Kesselmeier
et al., 2002; Malhi, 2002).  BVOC concentrations in ambient air depend on several factors,
which include emission rates from vegetation, atmospheric transport and mixing, as well as the
chemical composition and oxidative state of the atmosphere, which determines the sink of these
species.  BVOCs are important in the formation of tropospheric ozone ($O_3$) and secondary
organic aerosols (SOA).  BVOCs in the troposphere react with the major oxidants in the
atmosphere, which include tropospheric $O_3$, hydroxyl radicals ($^{\bullet}OH$, referred to, from here on,
as OH for simplicity) and nitrate radicals ($NO_3^{\bullet}$, referred to, from here on, as $NO_3$ for simplicity)
(Atkinson and Arey, 2003a).  These oxidants strongly affect the concentrations of atmospheric
BVOCs (Lelieveld et al., 2008; Di Carlo et al., 2004).  BVOCs are also crucial in the formation
of the stabilised Criegee intermediate – a carbonyl oxide with two free-radical sites – or its
derivative (Mauldin III et al., 2012; Welz et al., 2012), which also contributes to atmospheric
oxidation.  A complex range of reaction products are formed from atmospheric BVOCs,
including less volatile oxygenated compounds that condense to form aerosol particles.
Various studies have indicated the link between BVOCs and the formation of SOA (Vakkari et
al., 2015; Andreae and Crutzen, 1997; Ehn et al., 2014), while the influence of BVOCs on the
growth of newly formed aerosol particles has also been indicated (Kulmala et al., 2004; Tunved
et al., 2006).  However, there are many uncertainties associated with the exact chemical
reactions and physical processes involved in SOA formation and aerosol particle growth, which
largely depends on regional emissions and atmospheric processes (Kulmala et al., 2013; Ehn et
al., 2014).  Vakkari et al. (2015) indicated the importance of VOCs for new particle formation
and growth in clean background air in South Africa.  Therefore, it is essential to understand the
sources, transport and transformations of these compounds for air quality management and
climate change-related studies, as well as for the modelling of atmospheric chemistry at global,
regional and local scales (Laothawornkitkul et al., 2009; Peñuelas and Staudt, 2010; Peñuelas
and Llusià, 2003).





Long-term ambient BVOC measurements to establish seasonal cycles have been conducted
extensively in several regions, which include boreal forest (Hakola et al., 2009; Hakola et al.,
2000; Rinne et al., 2000; Rinne et al., 2005; Rantala et al., 2015; Räisänen et al., 2009;
Eerdekens et al., 2009; Lappalainen et al., 2009), hemiboreal mixed forest (Noe et al., 2012),
temperate (Spirig et al., 2005; Stroud et al., 2005; Fuentes et al., 2007; Mielke et al., 2010),
Mediterranean (Davison et al., 2009; Harrison et al., 2001) and tropical (Rinne et al., 2002)
ecosystems. Shorter campaigns have also been conducted in Western and Central Africa, which
include several different studies in the framework of the African Monsoon Multidisciplinary
Analyses (AMMA) (Grant et al., 2008; Saxton et al., 2007) and the EXPeriment for the
REgional Sources and Sinks of Oxidants (EXPRESSO) (Serca et al., 2001). Zunckel et al.
(2007) and references therein indicated that limited research has been conducted on BVOC
emissions in southern Africa, which consisted mainly of short campaigns measuring BVOC
emission rates. Considering that BVOC emissions on a global scale are considered to be
significantly higher (ca. 10 times) than the emission of anthropogenic VOCs, it is very
important that longer-term BVOC measurements are conducted in southern Africa.
Furthermore, a large part of the land cover in South Africa consists of a grassland bioregion, as
indicated in Figure 1. Although it is considered that grasslands cover approximately one quarter
of the Earth's land surface, relatively few studies have been conducted on BVOC emissions
from grasslands, while there are no long-term BVOC studies reported for these landscapes
(Bamberger et al., 2011; Ruuskanen et al., 2011; Wang et al., 2012). Therefore, the aim of this
study was to quantify the ambient BVOC concentrations over different seasons at a regional
background site in South Africa. In addition, the objective was also to characterise their
seasonal patterns, as well as to relate BVOC concentrations measured in southern Africa to
levels in other regions in the world. According to the knowledge of the authors, this is the first
record of ambient BVOC concentrations covering a full seasonal cycle in southern African and
for a grassland bioregion anywhere in the world.
**Insert Figure 1**





## 2    Measurement location and methods
### 2.1    Site description
Measurements were conducted at the Welgegund measurement station (26.57°S, 26.94°E, 1480
m a.s.l.) (Welgegund measurement station, 2016), which is located on the property of a
commercial maize and cattle farmer approximately 100 km west of Johannesburg, as indicated
in Figure 1. Welgegund is a regional background station with no pollutant sources in close
proximity. The distances to the nearest blacktop road and nearest town are approximately 10
and 30 km, respectively. Welgegund is, however, affected by the major anthropogenic source
regions in the north-eastern interior of South Africa (as indicated by the major large point
sources in Figure 1), which also include the Johannesburg-Pretoria conurbation (Tiitta, et al.,
2014). From Figure 1, it is also evident that the western sector contains no major point sources
and can therefore be considered to be representative of a relatively clean regional background.
Welgegund is geographically located within the South African Highveld, which is characterised
by two distinct seasonal periods, i.e. a dry season from May to September that predominantly
coincides with winter (June to August), and a wet season during the warmer months from
October to April. The dry period is characterised by low relative humidity, while the wet season
is associated with higher relative humidity and frequent rains that predominantly occur in the
form of thunderstorms. The mean annual precipitation is approximately 500 mm with
approximately >80 % of rain events occurring during the wet season. During the sampling
period, the coldest temperature recorded in winter at Welgegund was -5.1 °C in June 2011,
while the highest temperature recorded in summer was +35.6 °C in October 2011. The mean
maximum temperature ranges between 16 and 32 °C, while the mean minimum temperature
ranges between 0 and 15 °C. Winters are also characterised by frequent and severe frost days
(26-37 days per year) (Mucina and Rutherford, 2006).
### 2.2    Vegetation
The Welgegund measurement station is located in the Grassland Biome (Figure 1), which
covers 28 % of South Africa's land surface (Mucina and Rutherford, 2006). This biome has
been significantly transformed, primarily as a result of cultivation, plantation forestry,
urbanisation and mining (Daemane et al., 2010 and references therein). It has also been severely
degraded by erosion and agricultural development. The station is situated within Vaal-Vet



Sandy Grassland with Andesite Mountain Bushveld of the Savannah Biome prominent on
nearby ridges.  At present, only 0.3 % of the Vaal-Vet Sandy Grassland is statutorily conserved,
while the rest is mostly used for grazing and crop production.  In Figure 2, a land cover map
within a 60 km radius from Welgegund is presented indicating the extent of cultivation in this
region.  The land cover survey was performed within a region that was estimated to represent
the BVOC footprint at Welgegund, which was calculated from typical atmospheric lifetimes
(Table 1) of the species measured and the general wind speed(s) (Figure 3) at Welgegund.  The
immediate area surrounding Welgegund is grazed by livestock, with the remaining area covered
by crop fields (mostly maize and to a lesser degree sunflower).  In the demarcated 60 km radius,
a further three vegetation units of the Dry Highveld Grassland Bioregion (Grassland Biome)
and another two of the Central Bushveld Bioregion (Savannah Biome) are also present.  In
addition, alluvial vegetation is found associated with major rivers and inland saline vegetation
in scattered salt pans.
**Insert Figure 2**
The study area comprises a highly variable landscape with scattered hills and sloping, slightly
irregular, undulating plains, which are dissected by prominent rocky ridges.  Soil in the
catchment area is heterogeneous and rocky, varying from sandy to clayey depending on the
underlying rock types, such as andesite, chert, dolomite, mudstone, quartzite, sandstone and
shale.
Land use within the surrounding area is divided into six major land cover types, i.e. cultivated
land, grasslands, mountainous areas, plantations, urban areas and water bodies, as indicated in
Figure 2.  Mountainous areas, grassland and water bodies (riparian areas) comprised many
different vegetation units.  The other homogenous areas were anthropogenically altered and no
longer representative of the surrounding natural vegetation.  The study area is characterised by
a grassland-woodland vegetation complex, dominated by various grass and woody species, and
recognised by the presence of non-native species in altered environments.
The most dominant woody species of the entire study area include the trees *Celtis africana*,
*Searsia pyroides*, *Vachellia karroo* and *Ziziphus mucronata*, and the thorny shrub *Asparagus*
*laricinus*.  Tree diversity increases where there are patches of deep sand, characterised by





*Gymnosporia buxifolia* and *Vachellia erioloba*, or in mountainous areas, where *Euclea*
*undulata*, *Grewia flava* and *Senegallia caffra* become most prominent. Woody vegetation
occurs sparsely in grasslands and when present is found on isolated ridges, including the small
trees *Pavetta zeyheri*, *Vangueria infausta* and *Zanthoxylum capense*. In anthropogenically
altered environments, native species decrease and introduced species dominate, such as
*Eucalyptus camaldulensis*, *Pinus roxburghiana* and *Populus canescens* in plantations; *Gleditsia*
*triacanthos*, *Pyracantha coccinea* and *Salix babylonica* along rivers and water bodies; and
*Celtis sinensis*, *Melia azedarach* and *Robinia pseudoacacia* in the urban footprint.
The most dominant species of the grass sward in the entire study area include *Cynodon*
*dactylon*, *Eragrostis chloromelas*, *Heteropogon contortus*, *Setaria sphacelata* and *Themeda*
*triandra*. The dry, western grassland (Vaal-Vet Sandy Grassland specifically) is characterised
by *Anthephora pubescens*, *Cymbopogon caesius*, *Digitaria argyrograpta*, *Elionurus muticus*
and *Eragrostis lehmanniana*, and the moist Rand Highveld Grassland in the south-east by
*Ctenium concinnum*, *Digitaria monodactyla*, *Monocymbium ceresiforme*, *Panicum natalense*
and *Trachypogon spicatus*. The north-eastern parts of the study area on dolomite are dominated
by *Brachiaria serrata*, *Digitaria tricholaenoides*, *Eragrostis racemosa* and *Loudetia simplex*.
**2.3   Measurement methods**
**2.3.1   BVOC measurements and analysis**
BVOC measurements were conducted for a period of more than two years through a 13-month
sampling campaign from February 2011 to February 2012 and a 15-month sampling campaign
from December 2013 to February 2015. Samples were collected twice a week for two hours
during daytime (11:00 to 13:00 local time, LT) and two hours during night-time (23:00 to 1:00
LT) on Tuesdays and Saturdays. Several previous studies have demonstrated that the maximum
emissions of isoprene and monoterpenes from vegetation occur around midday (Fuentes et al.,
2000; Kuhn et al., 2002). Understandably, the chosen sampling schedule, i.e. same days each
week and same hours of the day, was prone to some bias. As mentioned by Jaars et al. (2014),
considering the distance of the sampling site from the nearest town and logistical limitations
during the sampling campaigns, the sampling schedule applied was the most feasible option
that enabled the collection of data for more than two years. VOCs were sampled at a height of
2 m above ground level, with a 1.75 m long inlet. The first 1.25 m of the inlet was a stainless
steel tube (grade 304 or 316) and the second 0.5 m was Teflon. To prevent the degradation of



BVOC by $O_3$, the stainless steel part of the inlet was heated to 120 ºC using heating cables and
thermostats (Thermonic), thereby removing ozone from the sample stream (Hellén et al.,
2012a). At regular intervals, the efficiency of this $O_3$ removal was verified with an $O_3$ monitor.
VOCs were collected with stainless steel adsorbent tubes (6.3 mm ED x 90 mm, 5.5 mm ID)
packed with Tenax-TA and Carbotrap-B by using a constant flow type automated
programmable sampler. A detailed description of the sampling procedure is presented by Jaars
et al. (2014). In short, the flow rate of the pump was set at between 100 and 110 ml $min^{-1}$
throughout the campaigns and was calibrated each week. Prior to sampling, all adsorbent tubes
were tested for leaks and preconditioned with helium for 30 minutes at 350 ºC at a flow of 40
ml $min^{-1}$.
Individual BVOCs were identified and quantified using a thermal desorption instrument
(Perkin-Elmer TurboMatrixTM 650, Waltham, USA) connected to a gas chromatograph
(Perkin-Elmer® Clarus® 600, Waltham, USA) with a DB-5MS (60 m, 0.25 mm, 1 µm) column
and a mass selective detector (Perkin-Elmer® Clarus® 600T, Waltham, USA). Samples were
analysed using the selected ion mode (SIM). A five-point calibration was performed by using
liquid standards in methanol solutions. Standard solutions were injected onto adsorbent tubes
that were flushed with helium at a flow of 100 ml $min^{-1}$ for 10 min in order to remove methanol.
BVOCs quantified for the two campaigns included isoprene with method detection limit (MDL)
between 1.2 and 2.4 pptv and for 2-methyl-3-butene-2-ol (MBO) between 0.9 and 1.4 pptv.
The monoterpenes (MT) (α-pinene, camphene, β-pinene, $\Delta^3$-carene, p-cymene, limonene, 1,8-
cineol, terpinolene, 4-acetyl-1-methylcyclohexene (AMCH), nopinone, bornylacetate and 4-
allylanisole) MDL was between 0.6 and 1.6 pptv. The sesquiterpenes (SQT) (longicyclene, iso-
longifolene, aromadendrene, α-humulene and alloaromadendrene) MDL was ~0.6 pptv. Since
the analytical system did not separate myrcene and β-pinene, β-pinene concentrations
determined were the sum of these two species. VOC concentrations were field and lab blank
corrected. When monthly median BVOC concentrations were calculated, sample
concentrations below the method detection limit (MDL) were replaced with ½MDL.

## 2.3.2  Ancillary measurements

Ancillary measurements continuously performed at the Welgegund station were used to
interpret the measured BVOC concentrations. General meteorological parameters, i.e.
temperature (T), relative humidity (RH), wind speed and direction, and precipitation were





measured. Soil temperature and moisture at different depths (5 and 20 cm) were measured with
a PT-100 and Theta probe ML2x (Delta-T), respectively. Additional soil moisture information
was obtained with a 100 cm PR2 soil moisture profile probe (Delta-T). Direct photosynthetic
photon flux density (PPFD) between 400 and 700 nm was measured with a Kipp & Zonen
pyranometer (CMP 3 pyranometer, ISO 9060:1990 Second Class).
Trace gas measurements were performed utilising a Thermo-Electron 43S sulphur dioxide
($SO_2$) analyser (Thermo Fisher Scientific Inc., Yokohama-shi, Japan), a Teledyne 200AU
nitrogen oxide ($NO_x$) analyser (Advanced Pollution Instrumentation Inc., San Diego, Cam
USA), an Environment SA 41M $O_3$ analyser (Environment SA, Poissy, France) and a Horiba
APMA-360 carbon monoxide (CO) analyser (Horiba, Kyoto, Japan). The net ecosystem
exchange (NEE) of carbon dioxide ($CO_2$) was measured with the eddy covariance method with
a Licor 7000 closed path infrared gas analyser (IRGA) and a three-dimensional Metek sonic
anemometer at a height of 9 m, which is well above the average tree height of 2.5 m (Räsänen
et al., 2016). A more detailed description of additional parameters monitored at Welgegund is
given by Beukes et al. (2015).

### 2.3.3 Lifetime of BVOCs

In Table 1, the atmospheric lifetimes ($\tau$) of BVOCs measured in this study calculated from OH-
and $O_3$ reactivity are reported. BVOC lifetimes according to $O_3$ reactivity were calculated with
Eq. (1):
$$\tau = \tau_{O3} = \frac{1}{k_{O3,}[\,O_3]} \tag{1}$$
where [$O_3$] is the annual average $O_3$ concentration (ca. 36 ppbv) measured during the two
campaigns at Welgegund and $k_{O3}$ the reaction rate constant for the reaction between a specific
BVOC and $O_3$. Since direct OH reactivity measurements were not available, the average
concentration of OH radicals ([OH]) (ca. 0.04 pptv) reported by Ciccioli et al. (2014) was used
in the calculations, using Eq. (2):
$$\tau = \tau_{OH} = \frac{1}{k_{OH,}[\,OH]} \tag{2}$$
where $k_{OH}$ is the reaction rate constant for the reaction between a specific BVOC and OH.





**Insert Table 1**
**3   Results and discussion**
**3.1   Meteorological conditions during the measurement campaigns**
Local meteorological influences on the measured BVOC concentrations are likely to be more
significant than regional impacts of air masses due to the short lifetimes associated with
atmospheric BVOCs (Table 1). Therefore, BVOC concentrations were only interpreted in
terms of local meteorological patterns and no back trajectory analyses were employed. In
Figure 3, the monthly medians of the meteorological parameters – precipitation, T, RH, wind
speed and -direction, and soil moisture depth (5 and 20 cm) – measured at Welgegund during
each of the two sampling campaigns are presented. From Figure 3a and b, the wet season
(October to April) associated with warmer months and the dry season (May to September)
associated with colder months as discussed in section 2.1 are evident. Rainfall in this region of
South Africa is typically characterised by relatively large inter-annual variability (Conradie et
al., 2016). The monthly median temperatures for the periods during which samples were
collected ranged between 8.8 and 13 °C in winter and 19.7 and 24.9 °C in summer (Figure 3b).
During the warmer months, temperatures up to 30 ºC and higher were reached frequently.
During the wet season, the monthly median RH ranged between 30 (with the onset of the wet
season) and 80 % (at the end of the wet season), while the RH ranged between 20 and 50 %
during the dry season (Figure 3c). The highest monthly median wind speeds occurred during
the warmer months (Figure 3d) when unstable meteorological conditions are prevalent in the
interior of South Africa (Tyson et al., 1996). The seasonal variations of wind direction during
the two sampling campaigns (Figure 3e) indicated that the prevailing wind direction was from
the northern to eastern sector, which agrees with the back trajectory analysis performed for the
first sampling period at Welgegund by Jaars et al. (2014). Soil moisture measurements
mimicked the seasonal precipitation pattern, i.e. higher soil moisture associated with the wet
season (Figure 3f and 3g). The soil moisture measurements conducted from January to August
at a depth of 20 cm were significantly higher during the first sampling campaign. During
December 2010 and January 2011, prior to the first sampling campaign, precipitation (Figure
3a) was clearly higher than during the second campaign, i.e. December 2013 to January 2014.
Subsequently, the soil moisture measured at 20 cm (Figure 3g) was clearly higher during the





first sampling campaign than during the second campaign from the beginning of the campaign
until the middle of the dry season.
**Insert Figure 3**
Figure 4 presents micrometeorological $CO_2$ flux measurements at Welgegund, which indicate
typical changes in the seasonal uptake of $CO_2$ by vegetation. Negative values (downward $CO_2$
flux) indicate the net uptake of $CO_2$ by vegetation, with the gross primary production (GPP)
exceeding the total respiration. Positive values indicate the emission of $CO_2$ by the vegetation.
A period of an approximately 0 (small positive) net $CO_2$ flux is observed in the winter months
that extend until September, which can be attributed to decreased microbial activity associated
with lower temperatures, low rainfall and most of the vegetation losing their leaves. The net
ecosystem exchange (NEE) at full light (maximum downward flux) increases gradually until
February in response to the increases of the photochemical efficiency of $CO_2$ assimilation in
the vegetation surrounding the site and the solar elevation angle. The daily maximum NEE
starts to decrease in March/April when the solar elevation angle declines and soil moisture
drops.
**Insert Figure 4**
**3.2   Contextualising BVOC concentrations measured at Welgegund**
In Table 2, the median (mean) and inter-quartile range (IQR, 25[th] to 75[th]) concentrations of the
BVOC species determined during the two sampling campaigns at Welgegund are presented. In
Table 3, the concentrations of BVOC species measured during other campaigns in South Africa
and the rest of the world are presented.
**Insert Table 2**
**Insert Table 3**



The most abundant species observed throughout the study was the monoterpene, α-pinene, and
the total monoterpene concentration was at least an order of magnitude higher compared to the
concentrations of other BVOC categories.  The total annual median (IQR) monoterpene
concentration was 120 (73-242) pptv during the first campaign and 83 (54-145) pptv during the
second campaign.  As indicated in Table 2, α-pinene, p-cymene and limonene were the
predominant compounds measured during the first campaign, constituting more than 63 % of
the ambient monoterpene concentrations, while during the second campaign, the dominant
monoterpenes were α-pinene, limonene and terpinolene, constituting more than 70 % of the
ambient monoterpene concentrations.  BVOC flux measurements conducted by Greenberg et
al. (2003) during SAFARI 2000 at a mopane woodland in Botswana indicated that 60 % of the
monoterpene flux was attributed to α-pinene, while limonene and β-pinene contributed almost
all of the rest of the monoterpenes.  Various studies in other regions have also indicated that α-
pinene is the dominant monoterpene in ambient air reflecting the ubiquitous nature of its
emission (Hellén et al., 2012b; Hakola et al., 2012; Noe et al., 2012).  During the AMMA
experiment, Saxton et al. (2007) also detected several monoterpenes in ambient air at Djougou
with concentrations generally higher than monoterpene concentrations recorded by Serca et al.
(2001) (less than 20 pptv) during EXPRESSO at a forest in Northern Congo.  Monoterpene
concentrations reported for boreal forest (Hakola et al., 2009; Hakola et al., 2000; Rinne et al.,
2000; Rinne et al., 2005; Rantala et al., 2015; Räisänen et al., 2009; Eerdekens et al., 2009;
Lappalainen et al., 2009), hemiboreal mixed forest (Noe et al., 2012), temperate (Spirig et al.,
2005; Stroud et al., 2005; Fuentes et al., 2007; Mielke et al., 2010), Mediterranean (Davison et
al., 2009; Harrison et al., 2001) and tropical (Rinne et al., 2002) ecosystems ranged between 40
and 7 200 pptv (Table 3).  Therefore, there is a large variation in the monoterpene
concentrations measured in different ecosystems, with concentrations measured at Welgegund
being in the low to mid-range.  Unlike isoprene that is approximately 10 times lower than
isoprene levels at other ecosystems in the world, the mean monoterpene concentration at
Welgegund is comparable to the previous studies at other ecosystems summarised in Table 3.
The annual median (IQR) isoprene concentration measured during the first campaign was 14 (6-
35) pptv, while the annual median (IQR) isoprene concentration measured during the second
sampling campaign was 14 (7-24) pptv.  The highest isoprene concentration, i.e. 202 pptv, was
recorded in summer (wet season).  Harley et al. (2003) reported that the maximum isoprene





concentration measured during an eight-day campaign in the wet season at a *Combretum-*
*Acacia* savannah in southern Africa was 860 pptv with a mean midday concentration of 390
pptv, which is considerably higher than isoprene levels measured at Welgegund. Ambient
BVOC measurements conducted by Saxton et al. (2007) at a rural site near Djougou, Benin in
June 2006 during the AMMA project indicated isoprene concentrations >3 000 pptv. Grant et
al. (2008) conducted VOC measurements at a small rural Senegalese village during September
2006 that was also a sampling location for the AMMA project and reported that isoprene, which
had a mean concentration of 300±100 pptv, was the only biogenic hydrocarbon present in all
air samples. Serca et al. (2001) reported ambient the mean isoprene concentration for a tropical
forest of Northern Congo during the EXPRESSO study to be 1820±870 pptv at the beginning
of the wet season and 730±480 pptv at the end of the wet season. Nakashima et al. (2014)
reported that the mean isoprene concentration at the Manitou Experimental Forest (MEF) was
$68 \pm 69$ pptv. In general, mean isoprene concentrations measured at Welgegund were at least
an order of magnitude smaller compared to other isoprene measurements in South Africa,
Africa and most other parts of the world.
The annual median (IQR) MBO concentrations measured during the first and second campaign
were 7 (3-16) and 4 (3-10) pptv, respectively. MBO and isoprene are both produced from
dimethylallyl diphosphate (DMADP) (Gray et al., 2011). Guenther (2013) indicated that MBO
is emitted from most isoprene emitting vegetation at an emission rate of ~1 % of that of
isoprene. However, MBO measured at Welgegund was approximately 30 % of the isoprene
concentrations, which indicated that the main source of MBO at Welgegund is not from
isoprene emitters, but from other MBO emitters. MBO concentration measurements at Manitou
Experimental Forest, USA were $1\,346 \pm 777$ pptv (Nakashima et al., 2014), which is three
orders of magnitude higher compared to the MBO levels measured at Welgegund. According
to the knowledge of the authors, there are no previous ambient MBO concentrations measured
for Africa.
Most SQTs are highly reactive species and are difficult to detect in ambient air samples, which
resulted in concentrations of these species being frequently below the detection limit of the
analytical procedure. This is also reflected in the concentrations of these species being an order
of magnitude lower compared to the other BVOC species measured in this study. The total
annual median (IQR) SQT concentration measured during the first sampling campaign was 8
(5-14) pptv and 4 (3-11) pptv during the second sampling campaign. The most abundant SQT





during the first sampling campaign was longicyclene with an annual mean concentration of 4
(1-4) pptv. During the second sampling campaign, α-humulene was the most abundant SQT
with an annual mean concentration of 3 (1-5) pptv.
The lower BVOC concentrations measured at Welgegund compared to other regions can mainly
be attributed to the much lower isoprene concentrations measured. However, monoterpenes that
are important for SOA formation are similar to levels thereof in other environments. In an
effort to explain the BVOC concentrations measured at Welgegund, a comprehensive
vegetation study was conducted, as described in section 2.2. The influence of the type of
vegetation in the region surrounding Welgegund on ambient BVOC concentrations will be
further explored.
Jaars et al. (2014) presented concentrations of aromatic VOCs measured at Welgegund during
the same two sampling campaigns discussed in this paper. The total BVOC concentrations
measured were at least an order of magnitude lower compared to concentrations of aromatic
VOCs measured at Welgegund. The most abundant aromatic compound, toluene, had a median
value of 630 pptv, while the most abundant BVOC measured, α-pinene, had a median value of
37 pptv. In addition, the median of the concentrations of the all the monoterpene species (120
and 83 pptv) was approximately six times lower compared to toluene concentrations (Jaars et
al., 2014).
**3.3  Seasonal variations**
In Figure 5, the panels on the left show monthly median concentrations of (a) isoprene, (b)
MBO, (c) monoterpenes and (d) SQT measured for the two campaigns, while the panels on the
right present the wet (October to April) and dry (May to September) season concentrations of
the respective compounds measured for the two campaigns. Seasonal variations in BVOC
concentrations are expected due to the response of emissions to changes in environmental
conditions, e.g. temperature and rainfall, as discussed in section 3.1, and the associated biogenic
activity. In addition, BVOC emission is expected to be lower during the winter months (June
to August), since foliar densities rapidly decrease due to deciduous trees dropping their leaves
in winter (Otter et al., 2002). As expected, it is evident that the concentrations of all the BVOC
species, with the exception of the isoprene (Figure 5a), and SQT values (Figure 5d) measured
during the second sampling campaign, were higher in the wet season. The wet season also had
more occurrences of BVOC concentrations that were higher than the range of the box and





whisker plot (whiskers indicating ±2.7σ or 99.3 % coverage if the data have a normal
distribution). In an isoprene and monoterpene emissions modelling study for southern Africa
conducted by Otter et al. (2003), it was estimated that BVOC emissions will decrease by as
much as 85 % in the dry winter season for grassland and savanna regions. BVOC
concentrations measured in this study indicated much lower decreases from summer (December
to February) to winter (June to August), with isoprene and monoterpene decreasing by only 37
and 29 %, respectively during the first sampling campaign, while isoprene and monoterpene
decreased by only 42 and 23 %, respectively during the second sampling campaign. This can
partially be attributed to the significant transformation of this biome, as discussed in section
2.2, with large areas transformed to cultivated land, as indicated in Figure 2. In addition, the
study by Otter et al. (2003) was conducted for the entire southern African region.
**Insert Figure 5**
The monthly median isoprene concentrations (Figure 5a) measured during the first sampling
campaign indicated the expected seasonal pattern with higher isoprene concentrations
coinciding with the wet and warmer months, with the exception of April that had lower isoprene
concentrations. Surprisingly, during the second sampling campaign, there was no distinct
seasonal pattern observed. However, higher isoprene concentrations seem to coincide with
higher wind speeds (Figure 3d), which are observed for both sampling campaigns. This
indicates that the major sources of isoprene measured at Welgegund can be considered not to
be within close proximity. However, since oxidation products of isoprene (e.g. methyl vinyl
ketone, methacrolein) were not measured in this study, more distant sources of isoprene could
not be verified. It is evident from Figure 2 that the region in close proximity of Welgegund in
the south-western to north-eastern sector largely comprises cultivated land, while in the north-
eastern to south-western sector the predominant land coverage is grassland and natural
vegetation. It is expected that isoprene emissions from the cultivated land will be lower
compared to savanna grassland (Otter et al., 2003). Therefore, if Welgegund is more frequently
affected by winds from the south-western to north-eastern sector, higher wind speeds will
coincide with higher isoprene levels, since the savanna grassland fetch region is distant from
Welgegund and related to the approximately three-hour atmospheric lifetime of isoprene due to
OH radicals.





In Figure 6, the wind roses for the BVOCs species measured in this study are presented. It is
evident that the highest isoprene concentrations for the first sampling period were associated
with winds originating from the south to south-western sector, i.e. predominantly from the
grassland region in close proximity during the first sampling campaign resulting in a relatively
more distinct seasonal pattern for isoprene levels. During the second sampling campaign,
higher isoprene concentrations were associated with winds originating from the south-western
to the northern sector, i.e. from the cultivated land area. Therefore, isoprene concentrations
measured during the second sampling period coincided predominantly with stronger wind
speeds from more distant fetch regions.
**Insert Figure 6**
Distinct seasonal patterns are observed for MBO (Figure 5b) concentrations during both
sampling campaigns, i.e. higher MBO concentrations coinciding with wet warm months and
lower levels corresponding with dry cold months (Figure 3). The MBO concentrations also
corresponded to the seasonal $CO_2$ uptake (Figure 4). It is also evident from Figure 5b that MBO
concentrations during the wet season in the first sampling campaign were higher compared to
the second sampling campaign, especially from February to April 2011. As mentioned in
section 3.1, the soil moisture measured at a depth of 20 cm (Figure 3g) during the first sampling
campaign was significantly higher from February to August compared to the second sampling
campaign. Therefore, these increased MBO levels measured during the first sampling
campaign can be attributed to increased emissions from deep-rooted plants, e.g. shrubs and
trees. In addition to decreased biogenic activity in the dry winter, the conversion of MBO to
isoprene in the atmosphere could also lead to decreased MBO levels during this period. Jaoui
et al. (2012) reported that MBO conversion to isoprene increased by an order of magnitude
during dry conditions compared to humid conditions. This can also contribute to elevated
isoprene concentrations measured during the dry months at Welgegund (Figure 5a).
No distinct seasonal pattern is observed for monoterpene and SQT concentrations, with the
exception of significantly higher levels measured from February to April 2011 during the first
sampling campaign. These increased monoterpene and SQT concentrations can also be
attributed to the significantly higher soil moisture measured at a depth of 20 cm during the first
sampling campaign (Figure 3g), as observed for the MBO. The monoterpene and SQT





concentrations measured during the first sampling campaign were generally higher compared
to the second sampling campaign. Otter et al. (2002) also reported a more pronounced seasonal
pattern for isoprene compared to monoterpene emissions at the Nylsvley Nature Reserve, which
is approximately 200 km north-west from Welgegund.
**3.4   BVOC emissions from surrounding vegetation**
As discussed in section 2.2 and indicated in Figure 2, Welgegund is situated in a region that has
been significantly transformed through cultivation. Cultivated land within the demarcated 60
km radius (Figure 2) consists mainly of maize and, to a lesser degree, sunflower production.
These cultivated lands are also typically characterised by eucalyptus trees, which have a very
high BVOC emission potential (Kesselmeier and Staudt, 1999), planted on their peripheries as
is evident in Figure 2. The grassland region in close proximity of Welgegund (south-western
to north-eastern sector) has a high diversity of grass and woody species, as mentioned in section
2.2. In general, it can be considered that the woody species in the grasslands are major sources
of all the BVOCs measured in this study. Otter et al. (2003) also considered woody vegetation
to be the most important in terms of BVOC emissions in southern Africa. It is generally
considered that crops and grass have very low isoprene-emitting capacities (Kesselmeier and
Staudt, 1999; Guenther, 2013). However, Schuh et al. (1997) indicate that sunflowers emit
isoprene; the monoterpenes α-pinene, β-pinene, sabinene, 3-carene and limonene; and the
sesquiterpene β-caryophyllene predominantly. In addition, Chang et al. (2014) (with references
therein) also indicated that isoprene has anthropogenic sources in urban areas, which indicates
that the surrounding towns can also contribute to the isoprene concentrations.
In an effort to determine possible sources of BVOC species concentrations, roses were
compiled, as presented in Figure 6. In general, the concentration roses indicated that isoprene
concentrations were higher form the western direction (indicated by the average and highest
concentrations), while wind direction did not indicate any significant differences in the
concentrations of the other BVOC species. On occasion, higher MBO, monoterpene and SQT
concentrations were observed from the south-eastern region, which may be attributed to a large
eucalyptus plantation approximately 15 km south-east from Welgegund, indicated in Figure 2.
However, high isoprene emissions are also usually associated with eucalyptus trees, which are
not observed in the isoprene concentration roses. Therefore, other sources of MBO,




monoterpene and SQT in these regions are most likely to be the main sources, which can
possibly include the urban footprint indicated in this region.
The similar concentration roses determined for monoterpenes and SQT during the first sampling
campaign can be attributed to similar sources of these species. However, most SQTs have short
atmospheric lifetimes (< 4 min) (Atkinson and Arey, 2003a), which indicated similar sources
within close proximity (~1 – 2 km radius) of Welgegund. Gouinguené and Turlings (2002)
indicated the emissions of several SQT from young maize plants by testing the effects of soil
humidity, air humidity, temperature, light and fertilisation rate on the emission of BVOCs from
these plants. Therefore, maize production may be a source of monoterpenes and SQT. The
higher SQT concentrations in the south-west and north-west can most likely be attributed to
smaller eucalyptus plantations within a 1 to 2 km radius, as indicated in Figure 2. The high
monoterpene concentrations determined during the second sampling campaign may be
associated with specific monoterpene emitting plants in the region.
Although a comprehensive vegetation survey has been conducted within a 60 km radius of
Welgegund, vegetation types have been identified only generally at this stage, as indicated in
section 2.2. Therefore, the predominant woody species in each of the regions surrounding
Welgegund associated with specific BVOC emissions have not yet been characterised.
**3.5   Statistical correlations**
Spearman's correlation analyses were applied to correlate the measured concentrations of
isoprene, MBO, monoterpenes and SQT measured to each other in order to substantiate sources
of these species. These correlations for the two sampling campaigns are presented in Table 4,
with correlations in the wet seasons listed in the lower bottom (highlighted light blue) and
correlations in the dry season presented in the top right (highlighted light grey). It is evident
that MBO had good correlations with monoterpenes and SQT in the wet season, as well as with
monoterpenes in the dry season during the first sampling campaign. Although not as distinct
as during the first sampling campaign, MBO did also correlate with monoterpenes during the
wet and dry season, as well as with SQT in the dry season during the second sampling campaign.
During the first sampling campaign, monoterpenes had a strong correlation with SQT in the
wet season and moderate correlation during the dry season, while strong correlations between
monoterpene and SQT were determined in the dry season and a moderate correlation during the
wet season during the second sampling campaign. As indicated previously, concentration roses





did indicate similar sources of MT and SQT, especially during the first sampling campaign,
which is signified by these correlations.
**Insert Table 4**
Spearman correlations between BVOCs and other paramaters measured at Welgegund did not
show significant correlations. However, in certain instances, good correlations were observed
between soil moisture and MBO, monoterpenes and SQT concentrations. This is expected,
since the monthly average concentrations of these species indicated increased levels thereof
that were associated with increased soil moisture from February to April 2011.  Therefore, in
an effort to further statistically explore the dataset, explorative multilinear regression was
performed by using all ancillary measurements as input data in order to indicate parameter
interdependencies on the BVOC concentrations measured.  In Figure 7, the root mean square
error (RMSE) difference between the calculated and measured BVOC concentrations, as a
function of the number of independent variables included in the optimum MLR solution, is
presented.  It is evident that interdependence between temperature, soil temperatures and PAR
yielded the largest decrease in RMSE for isoprene concentrations measured.  However, for
MBO, monoterpenes and SQT, a much more significant contribution from soil moisture is
observed to decrease the RMSE differences between calculated and measured BVOC levels. It
is also evident that the interdependence between soil moisture and soil temperature at 20 cm is
important to estimate MBO, monoterpene and SQT concentrations. Therefore, explorative
MLR indicated that temperature had the largest influence on isoprene concentrations, while soil
moisture was the most significant for MBO, monoterpenes and SQT levels.
**Insert Figure 7**
**3.6   Reactivity of BVOCs**
It is important to evaluate the significance of BVOCs on their atmospheric reactivity, since
these species are important precursor species in the photochemical formation of tropospheric
$O_3$ and SOA. This is particularly relevant for South Africa, with various recent studies




indicating that $O_3$ is currently the most problematic pollutant in South Africa (Laakso et al.,
2013; Venter et al., 2012; Beukes et al., 2013).  In addition, Vakkari et al. (2015) also indicated
the importance of VOCs for new particle formation and growth. Therefore, the $O_3$ formation
potential (OFP), reaction rates with $O_3$ and OH reactivities of the BVOCs measured in this
study were evaluated.
The OFP of BVOCs was determined by calculating the product of the average concentration
and the maximum incremental reactivity (MIR) coefficient of each compound, i.e. OFP =
VOC×MIR (Carter, 2009).   The MIR scale has been used to assess OFP for aromatic
hydrocarbons in numerous previous studies (Hoque et al., 2008; Jaars et al., 2014; Na et al.,
2005).  The reaction rates for reactions between $O_3$ and BVOCs were calculated with Eq. (3):
$\text{reaction rates} = k_{X,O3}[X][O_3],$ (3)
where [X] is the BVOC concentration, [$O_3$] the ozone concentration and $k_{X,O3}$ the reaction rate
constant for the reaction between X and $O_3$.  Since direct OH reactivity measurements were not
available, the OH reactivities ($s^{-1}$) of the BVOCs were calculated, using Eq. (4):
$\text{OH reactivity} = k_{X,OH}[X]$ (4)
where [X] is the BVOC concentration and $k_{X,OH}$ the reaction rate constant of the reaction
between X and OH.  In Table 5, the OFP calculated for each of the BVOCs measured in this
study, as well as the reaction rate constants for the reactions of these species with $O_3$ and OH,
are listed.
**Insert Table 5**
Table 5 indicates that, according to the OFP calculated with MIR coefficients, α-pinene,
isoprene and p-cymene had the highest OFP in descending order during the first sampling
campaign.  During the second sampling campaign, α-pinene also had the highest OFP, while
limonene and isoprene had the second and third highest OFPs, respectively.  A comparison of
the OFP calculated in this study to the OFP calculated by Jaars et al. (2014) for anthropogenic
aromatic hydrocarbons measured at Welgegund (with MIR coefficients) indicates that the OFP
of BVOCs is an order of magnitude smaller than the OFP of aromatic hydrocarbons at



Welgegund. The combined $O_3$ formation potentials of all the BVOCs measured calculated with
MIR coefficients during the first and second campaign were 1162 and 1022 pptv, respectively.
In Figure 8 (a), the monthly mean reaction rates for the reactions between $O_3$ and BVOCs
measured in this study are presented. Higher reaction rates between BVOCs and $O_3$ contribute
to increased atmospheric $O_3$ depletion. Significantly higher reaction rates were calculated for
February 2015. It is evident from Figure 8(a) that α-pinene and limonene had the highest
reaction rates with $O_3$, while isoprene exhibited relatively small contributions the $O_3$ depletion.
The other BVOCs also had relatively low reaction rates for their reactions with $O_3$. In Figure
8(b), the relative monthly contributions of each of the BVOCs to the total OH-reactivity of
BVOCs are presented. It is evident that largest contributions to the OH-reactivity of BVOCs
measured at Welgegund are from limonene, α-pinene and terpinolene for all of the months
during both sampling campaigns. This is expected, since monoterpenes had the highest
atmospheric concentrations compared to the other BVOCs measured in this study. It is also
evident, especially during the first sampling campaign, that isoprene levels increased with the
onset of spring in September.
**Insert Figure 8**
**4   Conclusions**
The annual median concentrations of isoprene, MBO, monoterpenes and SQT during the first
campaign were 14, 7, 120 and 8 pptv, respectively, and 14, 4, 83 and 4 pptv, respectively, during
the second campaign. The concentrations of BVOCs measured during the second campaign
were generally lower compared to levels during the first campaign, which can be attributed to
significantly higher rainfall during the wet season preceding the first campaign. The sum of
the concentration of the monoterpenes was an order of magnitude higher than the concentrations
of other BVOC species during both sampling campaigns, with α-pinene being the most
abundant species. Very low isoprene concentrations at Welgegund led to a significantly lower
total BVOC concentration compared to levels measured at other regions in the world and during
the SAFARI 2000 campaign in a South African national park. However, monoterpene
concentrations were similar to levels reported in most previous studies. In addition, total BVOC





concentrations were an order of magnitude lower compared to the total aromatic VOC
concentrations measured by Jaars et al. (2014) at Welgegund.
The monthly median MBO levels measured during both campaigns, as well as, although less
pronounced, the monthly median isoprene concentrations measured during the first campaign,
indicated a distinct seasonal pattern with higher MBO and isoprene concentrations coinciding
with the wet and warmer months. During the second campaign, higher isoprene concentrations
were associated with higher wind speeds, which were attributed to a larger distant source region.
No distinct seasonal pattern was observed for monoterpene and SQT concentrations, with the
exception of significantly higher levels measured from February to April 2011 during the first
campaign. In addition, MBO concentrations measured during these months were also
significantly higher. These increased MBO, monoterpene and SQT concentrations were
attributed to the significantly higher soil moisture measured at a depth of 20 cm resulting from
the wet season preceding the first campaign, which is indicative of biogenic emissions from
deep-rooted plants.
Concentration roses indicated that isoprene concentrations were higher from the western
direction, while wind direction did not indicate any significant differences in the concentrations
of other BVOC species. Woody species in the grassland region were considered to be the main
sources of BVOCs measured, while sunflower and maize crops were also considered to be
potential sources for BVOCs in this region.
Multilinear regression analysis utilising all the ancillary measurements at Welgegund indicated
that soil moisture had the most significant impact on atmospheric levels of MBO, monoterpenes
and SQT concentrations, while temperature had the greatest influence on isoprene levels.
The combined $O_3$ formation potentials of all the BVOCs measured calculated with MIR
coefficients during the first and second campaign were 1162 and 1022 pptv, respectively, with
isoprene and the monoterpenes: α-pinene, isoprene, p-cymene, limonene and terpinolene,
having the largest contribution to $O_3$ formation potential. α-Pinene and limonene had the
highest reaction rates with $O_3$, while isoprene exhibited relatively small contributions to the $O_3$
depletion. Limonene, α-pinene and terpinolene had the largest contributions to the OH-
reactivity of BVOCs measured at Welgegund for all of the months during both sampling
campaigns.





A comprehensive study on BVOC emissions from important plant species must be performed future studies in order to relate the emission capacities of vegetation types in the area surrounding Welgegund to the measured atmospheric BVOCs. It is also recommended that oxidation products of BVOC species are measured in future in order to verify distant source regions.

## Acknowledgements

The authors would like to acknowledge the Finnish Academy (project #132640), the University of Helsinki, the Finnish Meteorological Institute, the North-West University and the National Research Foundation (NRF) for financial support. Opinions expressed and conclusions arrived at are those of the authors and are not necessarily to be attributed to the NRF. Assistance with MATLAB from Ms Rosa Gierens is also acknowledged.

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

3    concentration measured for the two campaigns at Welgegund.

|  |  | $\tau_{OH}$ | $\tau_{O3}$ |
|---|---|---|---|
|  | Isoprene | 2.8 hr | 1 day |
|  | MBO | 10.3 hr | 7.5 day |
| Monoterpenes | α-Pinene | 5.3 hr | 3.6 hr |
|  | Camphene | 5.3 hr | 14.5 day |
|  | β-Pinene | 3.6 hr | 20.9 hr |
|  | $\Delta^3$-Carene | 3.2 hr | 8.5 hr |
|  | p-Cymene | 18.8 hr | 261.6 day |
|  | 1,8-Cineol | 12.5 hr | - |
|  | Limonene | 1.7 hr | 1.6 hr |
|  | Terpinolene | 12.6 hr | 2.3 hr |
|  | AMCH | 2.9 hr | - |
|  | Nopinene | 1.4 day | - |
|  | Bornylacetate | 1.5 day | - |
|  | 4-Allylanisole | 5.2 hr | 1.1 day |
| Sesquiterpenes | Longicyclene | 1.3 day | - |
|  | iso-Longifolene | 2.9 hr | 1.1 day |
|  | Aromadendrene | 4.5 hr | 1.1 day |
|  | α-Humulene | 1 hr | 21.6 min |



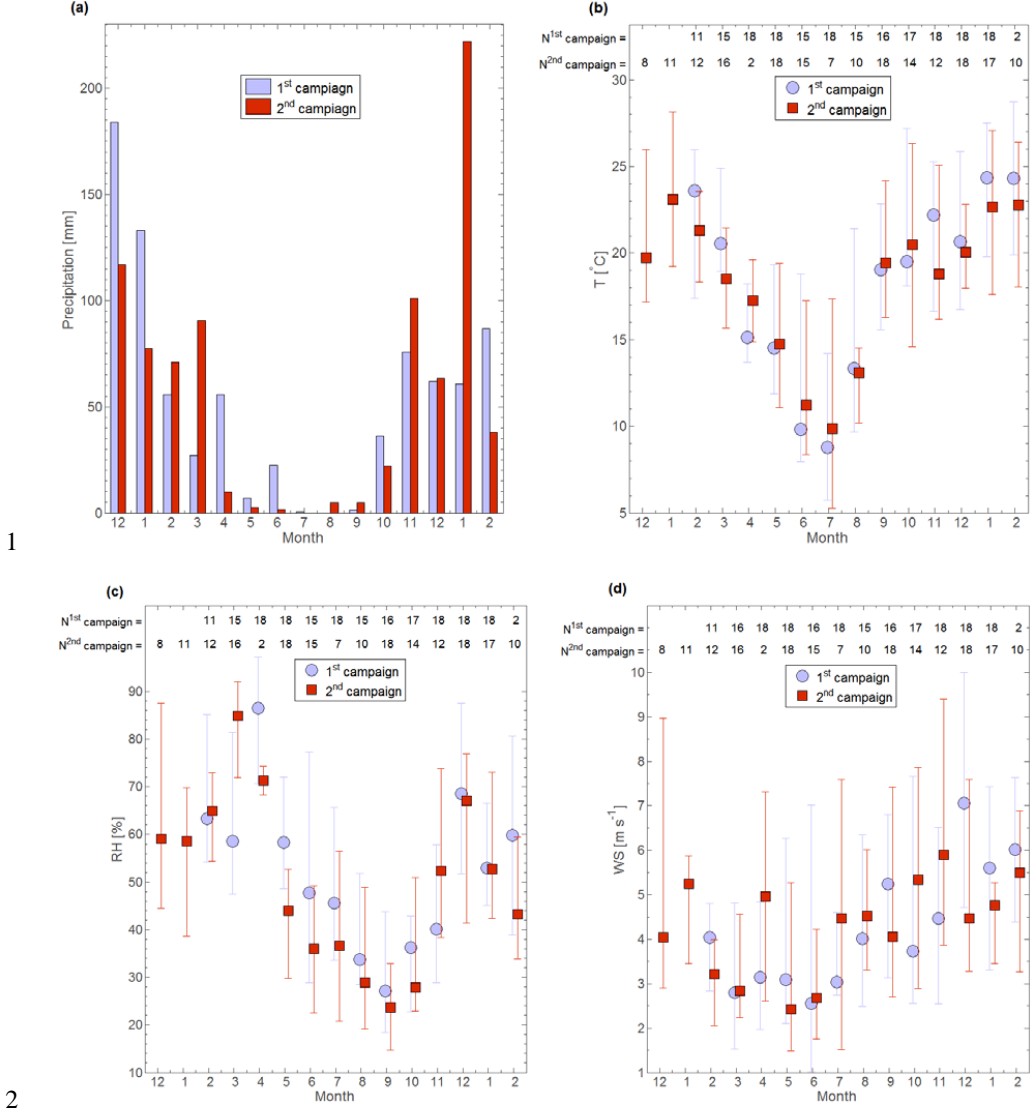

3  Figure 3.  Monthly variation of **(a)** accumulated precipitation, **(b)** temperature, **(c)** relative

4  humidity, **(d)** wind speed, **(e)** wind direction, and **(f)** and **(g)** soil moisture at 5 and 20 cm depth,

5  respectively.  Error bars indicate upper and lower quartiles.



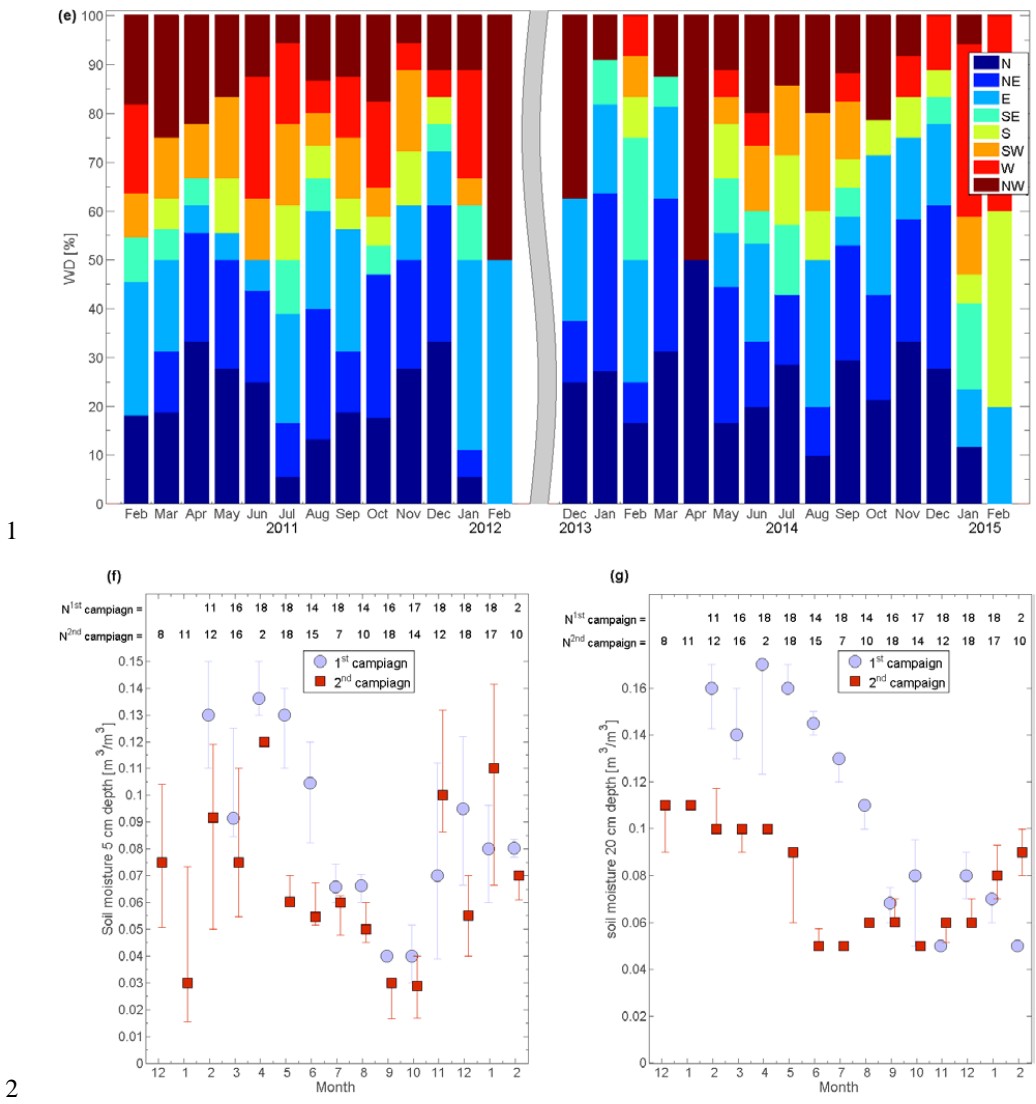

3      Figure 3.  Continued.





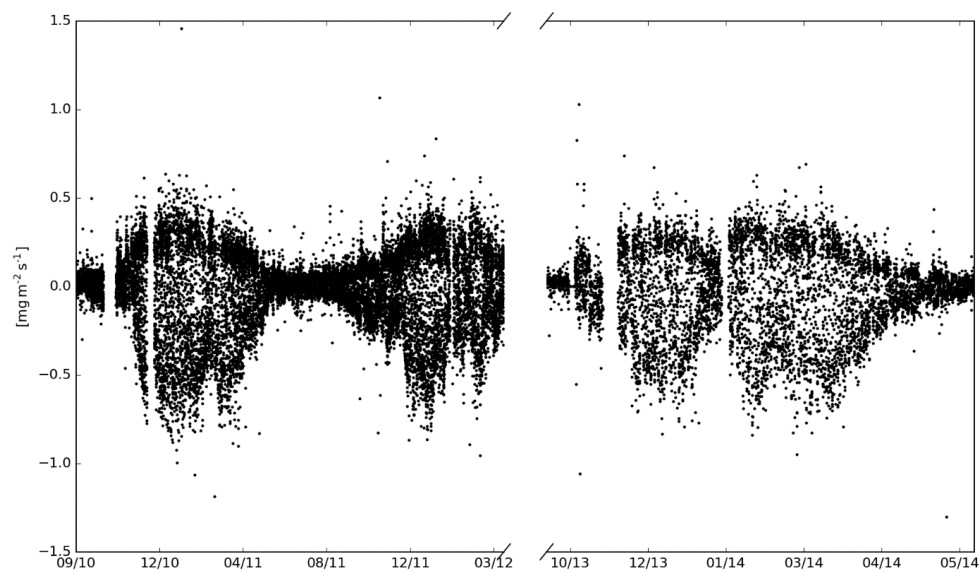

Figure 4. Micrometeorological $CO_2$ flux measurements at Welgegund (Räsänen et al., 2016).





1    Table 2. The ambient BVOC concentration for the two campaigns measured at Welgegund.

| pptv | First campaign | | | Second campaign | | |
|---|---|---|---|---|---|---|
| | Median (Mean) | IQR (25th - 75th) | N | Median (Mean) | IQR (25th - 75th) | N |
| Isoprene | 14 (28) | 6-35 | 187 | 14 (23) | 7-24 | 175 |
| MBO | 7 (12) | 3-16 | 178 | 4 (8) | 3-10 | 163 |
| Monoterpenes | | | | | | |
| α-Pinene | 37 (71) | 28-83 | 197 | 15 (57) | 9-23 | 191 |
| Camphene | 4 (8) | 2-9 | 178 | 2 (4) | 1-3 | 113 |
| β-Pinene | 9 (19) | 5-48 | 195 | 3 (5) | 2-6 | 171 |
| $\Delta^3$-Carene | 3 (6) | 2-5 | 156 | 2 (4) | 1-4 | 58 |
| p-Cymene | 20 (48) | 12-33 | 197 | 7 (15) | 4-13 | 186 |
| 1,8-Cineol | 3 (13) | 1-7 | 162 | 1 (2) | 1-2 | 75 |
| Limonene | 21 (30) | 9-40 | 197 | 16 (54) | 9-36 | 187 |
| Terpinolene | 4 (14) | 3-11 | 141 | 22 (28) | 16-34 | 25 |
| AMCH | 5 (7) | 1-12 | 41 | 3 (4) | 2-5 | 24 |
| Nopinene | 6 (7) | 4-9 | 167 | 8 (11) | 6-13 | 176 |
| Bornylacetate | 1 (2) | 1-2 | 49 | 2 (3) | 1-3 | 101 |
| 4-Allylanisole | 8 (11) | 5-13 | 118 | 1 (12) | 1-3 | 70 |
| Σ Monoterpenes | 120 (235) | 73-242 | | 83 (198) | 54-145 | |
| Sesquiterpenes | | | | | | |
| Longicyclene | 2 (4) | 1-4 | 152 | 1 (2) | 1-3 | 34 |
| iso-Longifolene | 2 (3) | 1-4 | 52 | 1 (1) | 1 | 7 |
| Aromadendrene | 1 (1) | 1 | 2 | 2 (2) | 1-3 | 73 |
| a-Humulene | 1 (1) | 1 | 3 | 1 (3) | 1-5 | 4 |
| Alloaromadendrene | 2 (3) | 1-4 | 31 | | | |
| Σ Sesquiterpenes | 8 (12) | 5-14 | | 4 (8) | 3-11 | |



1    Table 3.  Ambient BVOC concentrations (pptv) as reported by Noe et al. (2012) for various

2    ecosystems and then modified.  avg = mean value, med = median value, max = maximal value

3    of the measurements reported.

| Location | Isoprene | MBO | Monoterpenes | Date | References |
|---|---|---|---|---|---|
| **Grassland** | | | | | |
| Welgegund, SA | 28 (avg), 202(max) | 12 (avg), 61(max) | 235(avg), 1744(max) | Feb 2011-Feb 2012 | this study |
| | 23(avg), 182(max) | 7 (avg), 47(max) | 198(avg), 3081(max) | Dec 2013-Feb 2015 | this study |
| **Savannah** | | | | | |
| KNP, SA | 390(avg),860(max) | | | Feb 2001 | Harley et al. (2003) |
| Benin | >3000(max) | | >5000(max) | 7-13 Jun 2006 | Saxton et al. (2007) |
| Village, Senegal | 300(avg) | | | Sept. 2006 | Grant et al. (2008) |
| **Boreal** | | | | | |
| Hyytiälä, Finland | | | 900(avg), 1800(max) | 2000-2007 | Hakola et al. (2009) |
| | 40–110 | | 100–700 | Apr 2005 | Eerdekens et al. (2009) |
| | 220(med),360(max) | | 300(med), 600(max) | Summer 2006/2007 | Lappalainen et al. (2009) |
| | 70(med), 110(max) | | 200(med), 300(max) | Winter 2006/2007 | |
| | 110(avg), 430(max) | | 100(avg), 2700(max) | Jul 2004 | Rinne et al. (2005) |
| | | | 40–450 | 37 m, Aug 1998 | Rinne et al. (2000) |
| | | | 140–500 | 19.5 m, Aug 1998 | |
| | | | 450–630 | 2 m, Aug 1998 | |
| Huhus, Finland | | | 900(avg), 2160(max) | Jun.–Sep 2003 | Räisänen et al., (2009) |
| Pötsönvaara, Finland | 320–1690 | | 1700–3200 | Apr.–Oct 1997, 1998 | Hakola et al. (2000) |
| **Hemiboreal** | | | | | |
| Järvselja, Estonia | 360–2520 | | 1800–7200 | Spring and Summer 2010 | Noe et al. (2012) |
| | 120–200 (med) | | 400–1400 (med) | Oct 2009–Sep 2010 | Noe et al. (2012) |
| **Temperate** | | | | | |
| Michigan, USA | 2520(avg), 8160(max) | | 310(avg), 1100(max) | Summer 2008 | Mielke et al. (2010) |
| Jülich, Germany | 1980(avg), 10790(max) | | 250(avg), 1470(max) | Jul 2003 | Spirig et al. (2005) |
| Duke Forest, USA | 1500–2200 | | 310–790 | Jul 2003 | Stroud et al. (2005) |
| Oak Ridge, USA | 5000–15000 | | 500–1600 | Jul 1999 | Fuentes et al. (2007) |
| MEF, USA | 70(avg) | 1346(avg) | 0.497(avg) | 22-28 Aug. 2008 | Nakashima et al. (2014) |
| **Mediterranean** | | | | | |
| Castelpoziano, Italy | 141–250 | | 100–200 | May–Jun 2007 | Davison et al. (2009) |
| AM, Greece | 1500(avg), 7900(max) | | 900(avg), 5000(max) | Jul–Aug 1997 | Harrison et al. (2001) |



| Tropical | | | | |
|---|---|---|---|---|
| FNT, Brazil | 2000(avg), 4000(max) | 50(avg), 130(max) | Jul 2000 | Rinne et al. (2002) |
| NNNP, NC | 1820±870 | | 16–24 Ma. 1996 | Serca et al. (2001) |
| | 730±480 | | 21 Nov–11 Dec 1996 | |

SA = South Africa
WA = West Africa
KNP = Kruger National Park
MEF = Manitou Experimental Forest
AM = Agrafa Mountains
FNT = Floresta Nacional do Tapajos
NNNP = Nouabale-Ndoki National Park
NC = Northern Congo





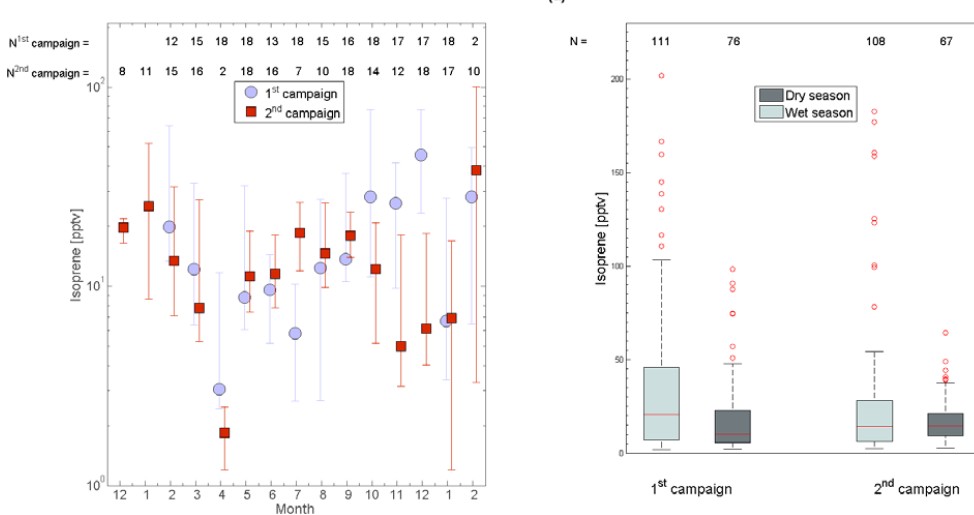

Figure 5. The panels on the left show monthly median concentrations of (a) isoprene, (b) MBO,
(c) monoterpenes and (d) SQT measured for the two campaigns. Error bars indicate upper and
lower quartiles. The values displayed near the top of the graphs indicate the number of samples
($N^{1st}$ and $N^{2nd}$ campaign) analysed for each month. The panels on the right show the wet and
dry season concentrations of the respective compounds measured for the two campaigns. The
red line of each box indicates the median (50th percentile), the black dot the mean, the top and
bottom edges of the box the 25th and 75th percentiles, the whiskers ±2.7σ or 99.3 % coverage if
the data have a normal distribution and the red circles outliers of the range of the box and
whisker plot. The values displayed near the top of the graphs indicate the number of samples
(N) analysed for the wet and dry season.



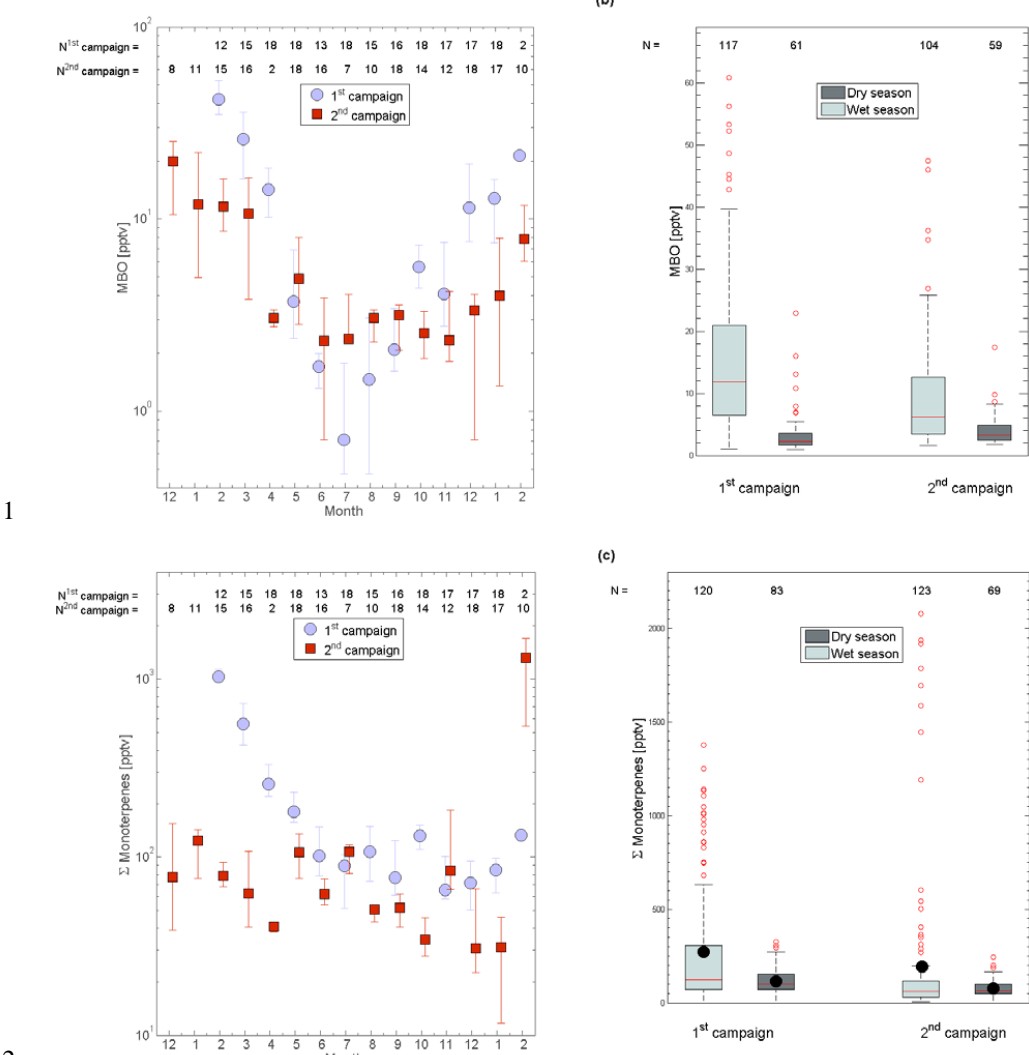

3    Figure 5.  Continued.





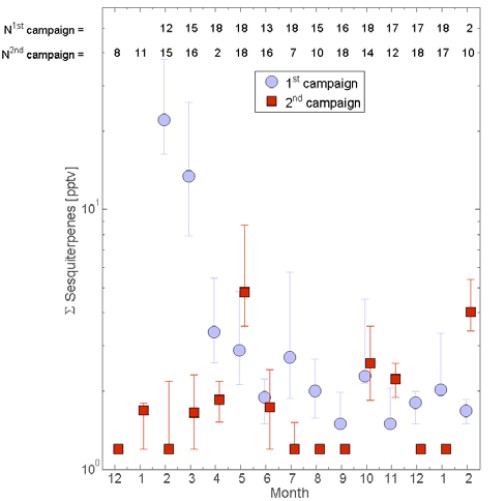

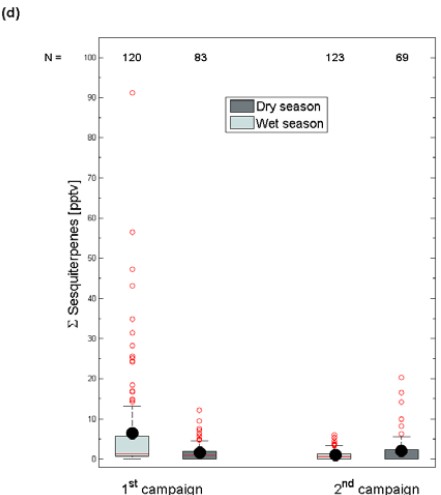

Figure 5. Continued.




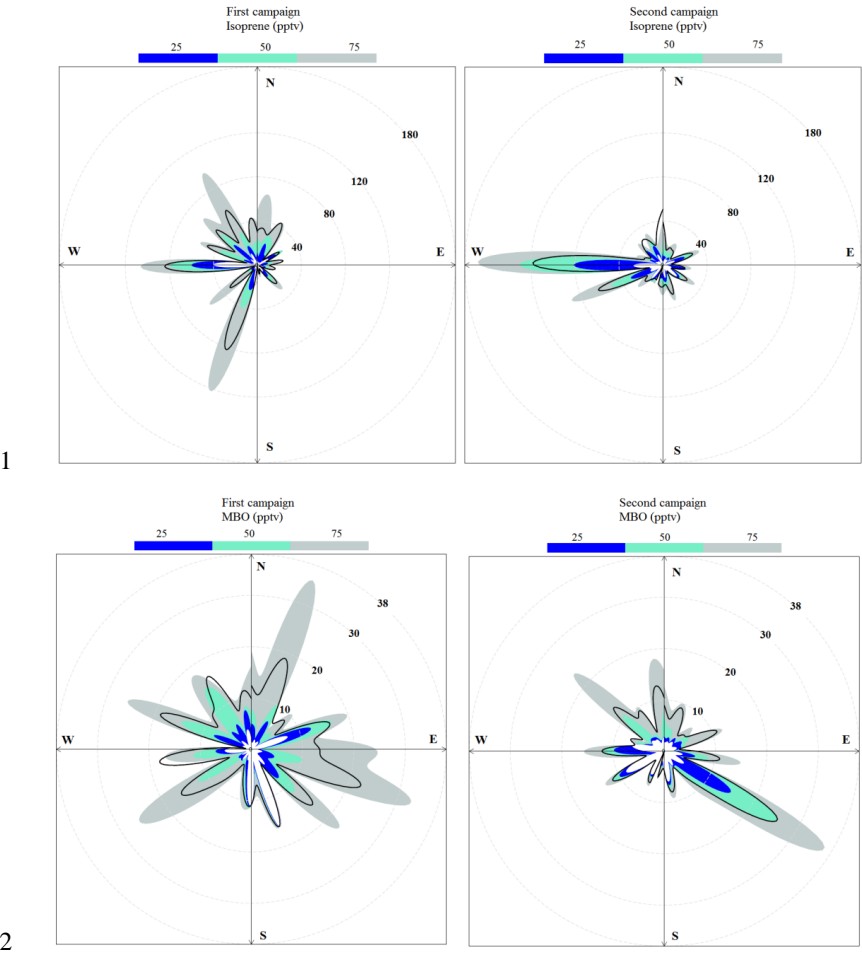

Figure 6. BVOC concentration rose at Welgegund for the two sampling campaigns. Different colours represent percentiles: blue 25 %, aquamarine 50 %, azure75 % and the black solid line the average.





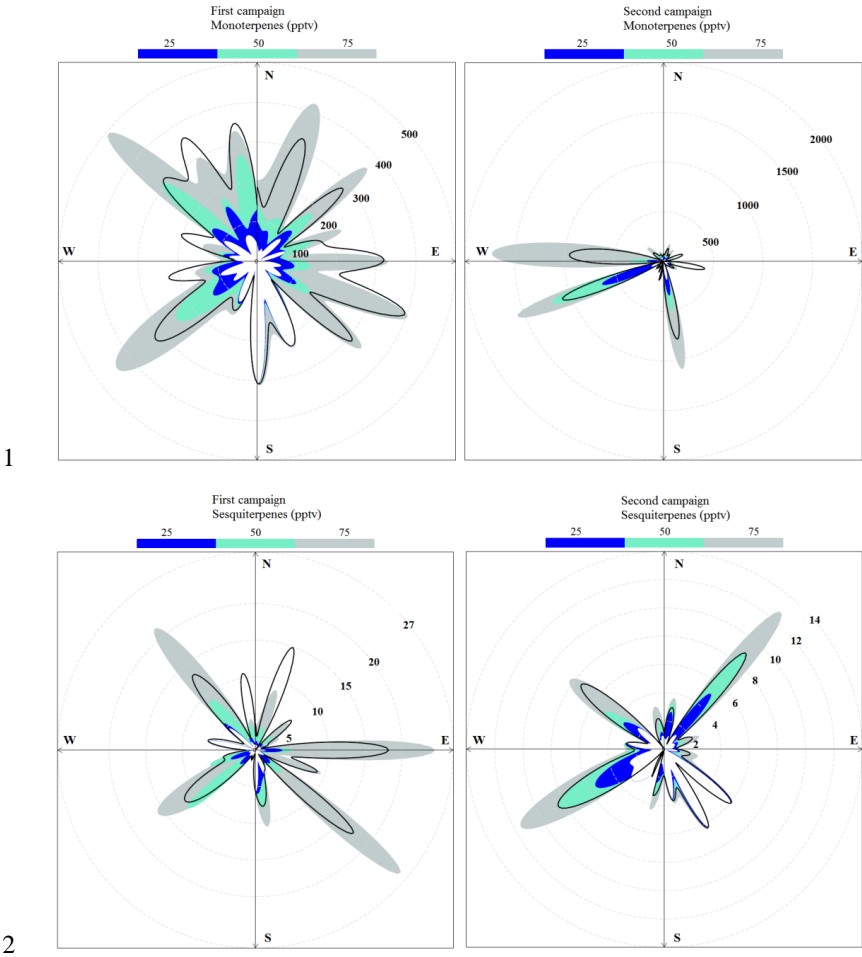

3    Figure 6. Continued.





Table 4. Spearman's correlation coefficients between the BVOCs during the wet and dry
season of the first campaign (a) and second campaign (b).
(a)

|  |  | Dry season | | | |
|---|---|---|---|---|---|
|  |  | Isoprene | MBO | MT | SQT |
| Wet season | Isoprene | - | **0.52** | **0.03** | **-0.10** |
| | MBO | 0.09 | - | **0.57** | **-0.10** |
| | MT | -0.20 | 0.68 | - | **0.27** |
| | SQT | -0.04 | 0.56 | 0.80 | - |

(b)

|  |  | Dry season | | | |
|---|---|---|---|---|---|
|  |  | Isoprene | MBO | MT | SQT |
| Wet season | Isoprene | - | **0.39** | **-0.11** | **0.09** |
| | MBO | 0.50 | - | **0.39** | **0.48** |
| | MT | 0.27 | 0.38 | - | **0.60** |
| | SQT | 0.20 | 0.01 | 0.26 | - |


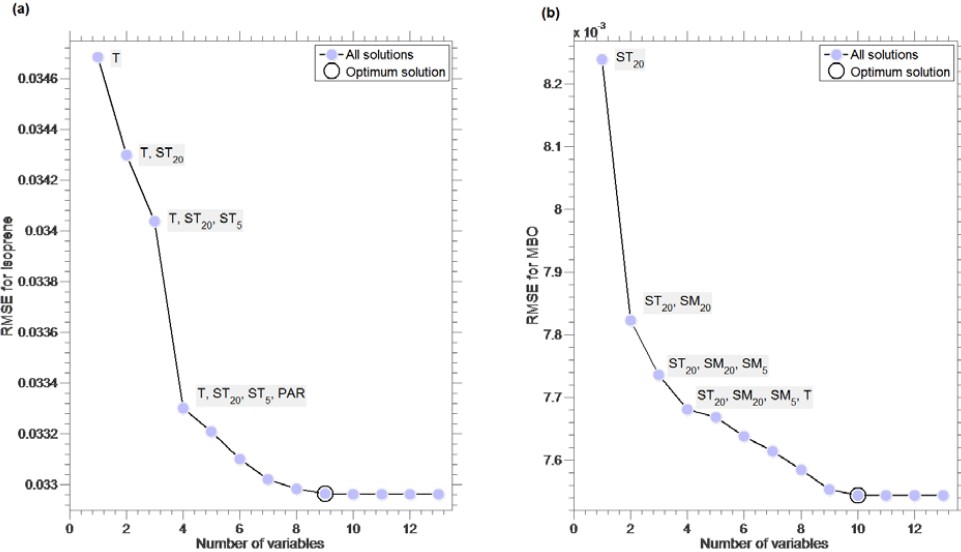

Figure 7.  The optimum combination of independent variables to include in a MLR equation to
calculate the dependant variable, i.e. BVOC concentrations.  The root mean square error
(RMSE) difference between the calculated and measured concentrations indicated that the
inclusion of (a) 9 parameters for isoprene, (b) 10 parameters for MBO, (c) 7 parameters for MT,
and (d) 12 parameters for SQT in the MLR solution was the optimum.

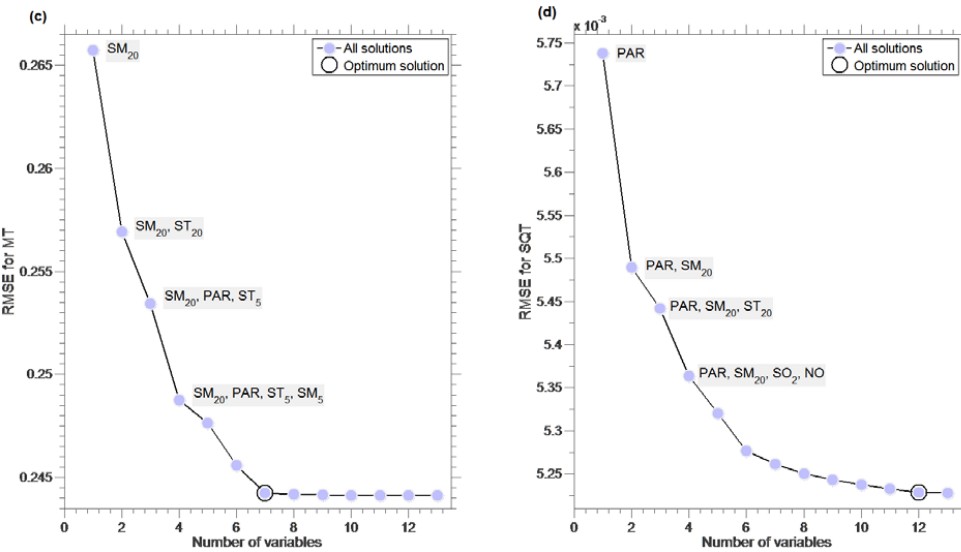

Figure 7.  Continued.





Table 5. Photochemical properties of measured BVOCs during the first and second campaign
at Welgegund.

| | | MIR[a] | First period | | Second period | | [cm³ molecule⁻¹ s⁻¹] | |
|---|---|---|---|---|---|---|---|---|
| | | | Avg | OFP | Avg | OFP | $k_{OH} \times 10^{12}$ | $k_{O3} \times 10^{18}$ |
| | Isoprene | 10.28 | 28 | 289 | 23 | 234 | 101.0 | 13.0 |
| | MBO | 4.73 | 12 | 56 | 7.7 | 37 | 27.5 | 1.8 |
| | α-Pinene | 4.38 | 71 | 313 | 57 | 251 | 53.7 | 86.6 |
| | Camphene | | 7.9 | | 3.8 | | 53.0 | 0.9 |
| | β-Pinene | 3.38 | 19 | 64 | 4.6 | 16 | 78.9 | 15.0 |
| | Δ³-Carene | 3.13 | 6.1 | 19 | 4.1 | 13 | 88.0 | 37.0 |
| | p-Cymene | 4.32 | 48 | 206 | 15 | 66 | 15.0 | 0.05 |
| | 1,8-Cineol | | 13 | | 1.9 | | 22.6 | |
| Monoterpenes | Limonene | 4.4 | 30 | 131 | 54 | 236 | 171.0 | 200.0 |
| | Terpinolene | 6.16 | 14 | 84 | 28 | 170 | 22.5 | 138.0 |
| | AMCH | | 6.7 | | 4.2 | | 98.6 | 430.0 |
| | Nopinene | | 7.3 | | 11 | | 8.6 | |
| | Bornylacetate | | 1.7 | | 3.1 | | 7.7 | |
| | 4-Allylanisole | | 11 | | 12 | | 54.3 | 12.0 |
| | Longicyclene | | 4.2 | | 1.7 | | 9.4 | |
| | iso-Longifolene | | 3.0 | | 0.9 | | 96.2 | 11.4 |
| Sesquiterpenes | Aromadendrene | | 1.0 | | 2.4 | | 62.5 | 12.0 |
| | α-Humulene | | 0.9 | | 2.7 | | 290.0 | 870.0 |
| | Alloaromadendrene | | 3.2 | | | | | |

[a]MIR denotes maximum incremental reactivity (g O₃/g VOCs) (Carter, 2009).
The rate constants are from Atkinson (2000) and Atkinson and Arey (2003b) except those for
α-humulene and longifolene OH reaction rates, which are from Shu and Atkinson (1995). Other
sesquiterpene data is from CSID:1406720, http://www.chemspider.com/Chemical-
Structure.1406720.html (last access: 2 May 2016). Predicted data is generated using the US
Environmental Protection Agency's EPI Suite.





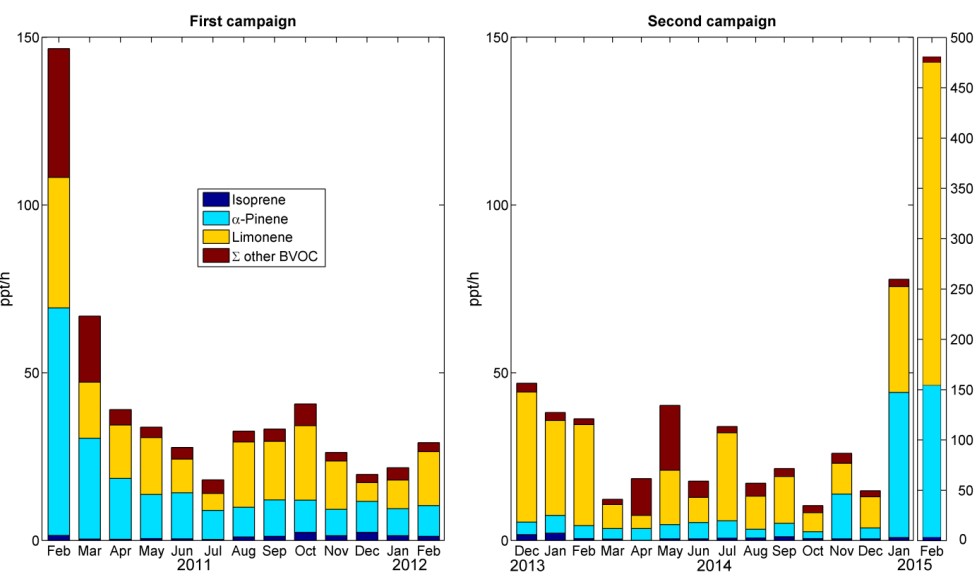

Figure 8a. Monthly means of reaction rates calculated for reactions between $O_3$ and BVOCs at

Welgegund. A secondary axis is introduced for reaction rates calculated for February 2015 due

to much higher reaction rates caluclated for this month.

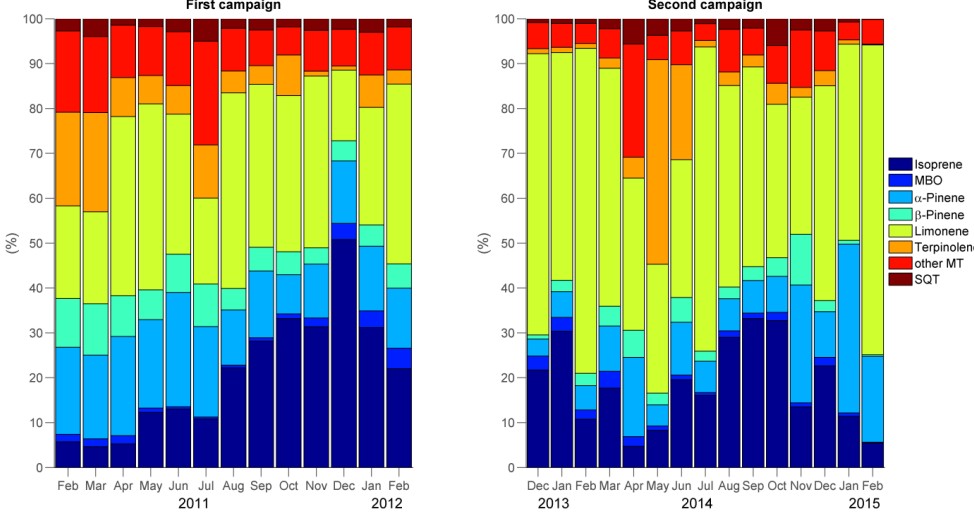

Figure 8b. The relative monthly contribution of different BVOCs to the OH-reactivity at

Welgegund.