# Peer review of "Measurements of biogenic volatile organic compounds at a"

_Atmospheric Chemistry and Physics, 2016_

## Referee Comment (RC1) · Anonymous Referee #1 · 12 Sep 2016

Jaars et al. report interesting measurements and analysis of biogenic VOC concentrations from an African grassland savannah ecosystem. The data from these regions are extremely limited and difficult to collect but are desired by the scientific community to understand the biological processes as well as the atmospheric abundance and fate of these molecules in these unexplored ecosystems. The paper should be an important reference and could inspire more research in those regions. Overall, I enjoyed reading the paper, thank you very much for this nice contribution, and I think the collected dataset is in itself extremely interesting so it deserves acceptation in ACP. However, I still feel the story has a significant potential for a little more in-depth analysis. In the relatively minor comments/questions below I just want to inspire some additional thoughts

[Figure]

and suggestions for further manuscript enhancement.

1) The paper is focused on the biogenic VOCs while the same authors described anthropogenic VOCs at Welgegund in a separate paper (Jaars et al. 2014) which maybe could be specified as a companion paper. I wonder if it could be interesting for the atmospheric chemistry context to try and look more closely at anthropogenic vs biogenic VOC interactions. For example, have you tried categorizing the data into pollution and clean periods based on high aromatics/NOx/SO2/O3 episodes to see for example if there is a difference in stress related monoterpenes or how different would be ozone and particle formation in these contrasting scenarios?

2) The paper suggests the concentrations of the biogenics were actually quite low compared to other woody biomass regions. Indeed, it could be very interesting to contrast this type of ecosystem to forests or tree plantations in Africa and elsewhere. One general issue is that the concentrations cannot tell us everything because despite the low concentration of a molecule there could still be a substantial flux and I was wondering if the authors have tried scaling these concentrations to turbulent parameters? In addition, isoprene concentration are known to exhibit strong diurnal variation as a function of time of day so there is implication of the sampling time (always the same time of day) at least for isoprene which warrants more dicussion. What percentage of isoprene concentration maximum was captured by these measurements could be easily inferred from a MEGAN algorithm for isoprene if the data for PAR and temperature are available.

3) The results of soil moisture relationship to monoterpene concentrations is very interesting. It would be instructive to see if the response was more like the threshold or did it exhibit a gradual dependence? It might be useful for potential modeling to see the actual scatter plots of soil moisture vs monoterpene concentration.

4) I was particularly intrigued by substantial concentrations of estragole (p-allylanisole). Unfortunately, this incredibly interesting aromatic compound is only listed in the tables

but I think it could be really nice to discuss this compound, in particular its likely origin (basil, anise, fennel, pines, palms?) and maybe even its behavior as a function of time/season. For example, I am wondering where it might be coming from and what its function is in this ecosystem. Could it be a pollinator attractant emitted by flowers (Misztal et al., 2010) or an insect deterrent emitted by conifers (Bouvier-Brown et al., 2009)? An additional minor suggestion would be to place p-allylanisole and p-cymene in a different category because these compounds are not strictly monoterpenes. You could consider something like "biogenic benzenoid" or "monoterpene-related BVOCs". AMCH is not strictly a monoterpene either but can be considered an oxygenated terpene.

5) Have you observed any monoterpenes (or other BVOCs) related to stress? For example $\beta$-ocimene, methyl salicylate, green leaf volatiles?

6) Could some emissions at the Welgegund site have floral origin which could further explain why isoprene is relatively low whereas monoterpenes (and potentially other compounds such as p-allylanisole) are relatively abundant? Is the flowering happening an entire year round or seasonally? It would be very interesting because floral BVOCs from meadow-like flowering ecosystems can sometimes be abundant but receive relatively little attention compared to foliar emissions.

7) Table 4 contains interesting correlations, in particular, that MBO correlates with monoterpenes. Are these compounds coming from a conifer-like sources? On the other hand, I wonder if the result of isoprene correlating with MBO is more unexpected and it is also not discussed. Baker et al. 2001 found that if MBO is thermally treated (as is the case in GC) it can dehydrate and be detected as isoprene. Do you think this could be the case? While this is not meant to be a criticism, and given the different wind-roses and dependence on soil moisture/temperature perhaps the issue was probably minor but I still think it is worth giving this potential issue a general thought and discuss implications for isoprene/MBO data interpretation.

8) Figure 4. This is an interesting figure. It seems that it is adopted from a different manuscript but clearly shows beautiful CO2 assimilation during the day and respired carbon during the night. If the data were available, it might be worth coloring these markers by PAR to better visualize assimilation vs respiration vs potentially anthropogenic CO2(?).

9) It is mentioned several times (abstract, P22 L15 and in other places) that isoprene concentrations were higher from the western direction. "western direction" is not very informative for a reader in particular in the abstract. Careful reading points to the sentence in P5 L11 that "...western sector contains no major point sources and can therefore be considered to be representative of a relatively clean regional background". Maybe you meant to say that this direction does not contain any *anthropogenic* point sources of isoprene? Otherwise I wonder where this isoprene is coming from? If isoprene concentrations exhibit temperature dependence, it implies biogenic source but if there is no vegetation to the west, could there be a different source (e.g. heated rubber?). My suggestion would be simply to expand more clearly on the potential sources of the western isoprene.

10) I understand the median is often used to represent more episode-free concentration scenario. However, isoprene is only emitted during the day, so does it still make sense to show the monthly median for isoprene? Because you were collecting data both during the day and at night, I think it could be very interesting to separate day and night concentrations. In particular because monoterpenes unlike isoprene can often accumulate during the night in a shallow boundary layer so the overall median (or mean) concentration differences between the compounds (e.g. isoprene and monoterpenes) may not reflect strictly their emission strengths or true variability. I wonder if looking at some of the episodic events of high concentrations would not be an even more interesting opportunity to understand the chemistry scenarios.

11) Further to the point above, you are talking about ozone formation potentials from these BVOCs but maybe it would be worthwile to show some oxidant data. I am just

curious how ozone (and also NOx, SO2, etc.) concentrations varied during the daytime and nighttime VOC sampling times and if it could teach us anything about the chemistry at the site.

12) It is recommended that the conclusions are made more succinct and emphasize major take-home messages which should be even more impressive than just the summary of the paper. For example, one could consider concluding about the implications for atmospheric chemistry and air quality in the region. In particular, the last two sentences of conclusions are unclear but the synopsis of future measurements is definitely needed to attract more attention and support more measurements in these almost completely unexplored regions of Africa.

References:

Baker, B., Guenther, A., Greenberg, J., and Fall, R.: Canopy Level Fluxes of 2-Methyl-3-buten-2-ol, Acetone, and Methanol by a Portable Relaxed Eddy Accumulation System, Environ Sci Technol, 35, 1701-1708, 10.1021/es001007j, 2001.

Bouvier-Brown, N. C., Goldstein, A. H., Worton, D. R., Matross, D. M., Gilman, J. B., Kuster, W. C., Welsh-Bon, D., Warneke, C., de Gouw, J. A., Cahill, T. M., and Holzinger, R.: Methyl chavicol: characterization of its biogenic emission rate, abundance, and oxidation products in the atmosphere, Atmos. Chem. Phys., 9, 2061-2074, 2009.

Misztal, P. K., Owen, S. M., Guenther, A. B., Rasmussen, R., Geron, C., Harley, P., Phillips, G. J., Ryan, A., Edwards, D. P., Hewitt, C. N., Nemitz, E., Siong, J., Heal, M. R., and Cape, J. N.: Large estragole fluxes from oil palms in Borneo, Atmos. Chem. Phys., 10, 4343-4358, 10.5194/acp-10-4343-2010, 2010.

---

## Referee Comment (RC2) · Anonymous Referee #2 · 21 Oct 2016

Summary: The authors present two separate years of atmospheric measurements of biogenic volatile organic compounds (BVOCs) at a grazed savannah-grassland-agriculture landscape in South Africa. This long-term and chemically detailed data set allows the authors to accurately assess the seasonal variability of these reactive compounds and provides the scientific community with a valuable dataset of BVOC emissions in a rarely studied ecosystem. The author's present a careful comparison of the observed emissions to the surrounding vegetation and other physical parameters such as soil moisture. The contributions of each species to potential ozone formation are explored. The data is of high quality and the manuscript is very well written. I suggest publication with only minor revisions.

Technical comments (P=page number and L=line number):

P1L22 and P3L5: These are not exactly the same statements. The first infers that anthropogenic sources only contribute 10% of the global annual VOC budget. The second statement (referencing Guenther et al.) states that 90% of BVOC emissions are from vegetation/terrestrial sources (i.e., only 10% from oceanic sources). Please be sure that these distinctions are more clear in the manuscript to avoid confusion by clarifying the contribution of BVOC vs. anthropogenic VOC. Also, do the emissions refer to mass or carbon or some other unit?

P2L12: lower for the grassland savannah or the other landscapes?

P4L25 and P29L2: African to Africa

P8L3: Can you state what the efficiency of the ozone removal was?

Section 2.3.1. Please add a brief description on how the samples were stored/transported prior to analysis. How much time would elapse between collection and analysis?

Section 3.5. How much of the correlation is simply driven by the fact that MBO, MTs, and SQTs generally have higher concentrations at night while isoprene will have the largest daytime emission? It would be interesting to compare the day and night time values of the compound classes, perhaps just for the wet season when emissions were enhanced.

---

## Author Comment (AC1) · 7 Nov 2016

Jaars et al. report interesting measurements and analysis of biogenic VOC concentrations from an African grassland savannah ecosystem. The data from these regions are extremely limited and difficult to collect but are desired by the scientific community to understand the biological processes as well as the atmospheric abundance and fate of these molecules in these unexplored ecosystems. The paper should be an important reference and could inspire more research in those regions. Overall, I enjoyed reading the paper, thank you very much for this nice contribution, and I think the collected dataset is in itself extremely interesting so it deserves acceptation in ACP. However, I still feel the story has a significant potential for a little more in-depth analysis. In the relatively minor comments/questions below I just want to inspire some additional thoughts and suggestions for further manuscript enhancement.

We would like to thank Referee #1 for the positive review of this paper and acknowledging the relevance of the work presented, as well as for indicating appreciation of the manuscript. We would also like to thank Referee #1 for the relatively minor comments/questions, which were each carefully considered in order to answer the questions raised or implement the suggestions made. Below is a point-by-point response to each of these comments/questions.

1) The paper is focused on the biogenic VOCs while the same authors described anthropogenic VOCs at Welgegund in a separate paper (Jaars et al. 2014) which maybe could be specified as a companion paper. I wonder if it could be interesting for the atmospheric chemistry context to try and look more closely at anthropogenic vs biogenic VOC interactions. For example, have you tried categorizing the data into pollution and clean periods based on high aromatics/NOx/SO2/O3 episodes to see for example if there is a difference in stress related monoterpenes or how different would be ozone and particle formation in these contrasting scenarios?

We agree with Referee #1 that this paper could be specified as a companion paper to the paper published on anthropogenic VOCs measured at Welgegund by Jaars et al., 2014. This will also contribute to enhancing the citability of these papers. We will follow this matter up with the handling editor.

We also agree that we must look at the anthropogenic vs biogenic interactions, which was recognised during preparation of the both the anthropogenic and biogenic VOC papers. Therefore a third paper was

prepared (and very close to submission) where positive matrix factorisation (PMF) analysis was performed in order to pull together the anthropogenic and biogenic VOCs measurements together with other ancillary measurements conducted at Welgegund e.g. $SO_2$, $NO_2$ and $O_3$. The $O_3$ formation potential relating to each of the factors determined with PMF will be explored in this paper. The PMF analysis will also be taken further in another paper where the reactions of all the VOCs (anthropogenic and biogenic) with oxidants, e.g. OH, $O_3$ will be further explored with atmospheric models in order to establish, for instance, whether $O_3$ formation within the region is VOC- or $NO_2$-limited. Therefore these two papers that are currently in preparation are included as future perspectives in the "Conclusions" section as follows:

"…BVOCs measured at Welgegund. In addition, the interactions between anthropogenic and biogenic VOCs must also be further explored, together with other ancillary measurements conducted at Welgegund (e.g. $SO_2$, $NO_2$ and $O_3$). Future work must also include investigating the reactions of all the VOCs measured with atmospheric oxidants (e.g. $^\bullet$OH and $O_3$) with atmospheric chemistry models in order to establish, for instance, whether $O_3$ formation within the region is VOC- or $NO_2$-limited."

In addition, as indicated in our response to Comment 5, no notable amounts of stress related monoterpenes were detected, which could therefore not be investigated.

2) The paper suggests the concentrations of the biogenics were actually quite low compared to other woody biomass regions. Indeed, it could be very interesting to contrast this type of ecosystem to forests or tree plantations in Africa and elsewhere. One general issue is that the concentrations cannot tell us everything because despite the low concentration of a molecule there could still be a substantial flux and I was wondering if the authors have tried scaling these concentrations to turbulent parameters? In addition, isoprene concentration are known to exhibit strong diurnal variation as a function of time of day so there is implication of the sampling time (always the same time of day) at least for isoprene which warrants more dicussion. What percentage of isoprene concentration maximum was captured by these measurements could be easily inferred from a MEGAN algorithm for isoprene if the data for PAR and temperature are available.

In our discussion in Section 3.2 and in Table 3 our BVOCs concentrations are compared to other BVOC measurements in Africa and other parts of the world, which include measurements at forests. We indicate that BVOC emissions were quite low, which is mainly attributed to the significant lower isoprene levels measured at Welgegund. Therefore we consider that this part of the comment is addressed in our manuscript.

We agree with Referee #1 that concentrations are influenced by factors other than fluxes and so a low concentration does not always indicate a low flux. In addition to fluxes, these VOC concentrations are

also sensitive to oxidation rates and boundary layer height. Greenberg et al. (1999) have shown that isoprene concentrations at most sites are relatively constant during daytime since the strong diurnal pattern of emission is offset by the strong diurnal variation in oxidation rate and boundary layer height. As suggested by the reviewer, we used the observed temperature and PAR at the site to drive the MEGAN BVOC emission model and found that the measurement time (11:00 to 13:00 local time) captured most of the period of maximum isoprene emission (typically about 12:00 to 2:00 local time). By assuming a typical diurnal variation in VOC oxidation rate and boundary layer height, we also find that the isoprene concentration of the measurement time is representative of the daytime isoprene concentration.

In addition to our response to this comment, the median (mean) daytime to night-time concentration ratios of all the BVOCs measured during both sampling campaigns were included in Table 2, in order to indicate the light dependence of BVOC emissions. These daytime to night-time concentration ratios indicated that there were not significant differences in levels of most of the BVOCs measured during daytime and night-time at Welgegund, with the exception of isoprene measured during the first sampling campaign, as well as two monoterpenes (terpinolene, bornylacetate) and the SQT aromadendrene measured duirng the second sampling campaign. Therefore isoprene levels measured during the first sampling campaign reflected the light dependency usually associated with isoprene emissions. However, daytime to night-time concentration ratios of isoprene did not exhibited the very strong light dependency typically associated with atmospheric isoprene concentrations, which could be attributed to the characteristics of sources of these species that are discussed later in the manuscript. Therefore the light dependency associated with these small number of BVOCs measured at Welgegund was addressed in the first paragraph in Section 3.2 as follows:

"In Table 2, the median (mean) and inter-quartile range (IQR, $25^{th}$ to $75^{th}$) concentrations, as well as the median (mean) daytime to night-time concentration ratios of the BVOC species determined during the two sampling campaigns at Welgegund are presented. It is evident from the median (mean) daytime to night-time concentration ratios that there were not significant differences in levels of most of the BVOCs measured during daytime and night-time at Welgegund, with the exception of isoprene measured during the first sampling campaign, as well as the monoterpenes terpinolene and bornylacetate, and the SQT aromadendrene measured duirng the second sampling campaign. Isoprene levels during the first sampling campaign were approximately two times higher during daytime, which reflect the light dependency usually associated with isoprene emissions. However, daytime to night-time concentration ratios of isoprene did not exhibited the strong light dependency typically associated with atmospheric isoprene concentrations, which could be attributed to the characteristics of sources of these species that are discussed in subsequent sections. The temperature and photoactive radiation (PAR) measurements at Welgegund was used in the MEGAN BVOC emission model, which indicated

that the measurement time (11:00 to 13:00 local time) captured most of the period of maximum isoprene emission (typically about 12:00 to 2:00 local time). In addition, by assuming a typical diurnal variation in VOC oxidation rate and boundary layer height, it was also found that the isoprene concentration of the measurement time is representative of the daytime isoprene concentration (Greenberg et al., 1999). In Table 3, the concentrations of BVOC species measured during other campaigns in South Africa and the rest of the world are presented."

Greenberg, J. P., A. Guenther, P. Zimmerman, W. Baugh, C. Geron, K. Davis, D. Helmig and L. F. Klinger (1999). "Tethered balloon measurements of biogenic VOCs in the atmospheric boundary layer." Atmospheric Environment **33**(6): 855-867.

3) The results of soil moisture relationship to monoterpene concentrations is very interesting. It would be instructive to see if the response was more like the threshold or did it exhibit a gradual dependence? It might be useful for potential modeling to see the actual scatter plots of soil moisture vs monoterpene concentration.

We agree with Referee #1 that this data could be useful for potential modelling. Therefore, scatter plots of soil moisture vs monoterpene and soil moisture vs SQT were included as supplementary material in the manuscript and are presented below:

(a)

[Figure]

[Figure]

(b)

[Figure]

Figure S1. Correlation between soil moisture and monoterpene concentrations (a), and soil moisture and SQT (b)

These plots clearly shows higher monoterpene and SQT concentrations associated with increased soil moisture. In addition, the following sentences were added to the text in Section 3.3:

"…monoterpene and SQT concentrations measured during the first sampling campaign were generally higher compared to the second sampling campaign.  In Figure S1 (a) and (b) the relationship between soil moisture and monoterpene concentrations, as well as between soil moisture and SQT are presented, respectively.  It is evident that higher concentrations of monoterpene and SQT are associated with higher soil moisture measured at a depth of 5 and 20 cm.  Otter et al. (2002) also reported a more pronounced seasonal pattern…"

4) I was particularly intrigued by substantial concentrations of estragole (p-allylanisole). Unfortunately, this incredibly interesting aromatic compound is only listed in the tables but I think it could be really nice to discuss this compound, in particular its likely origin (basil, anise, fennel, pines, palms?) and maybe even its behavior as a function of time/season. For example, I am wondering where it might be coming from and what its function is in this ecosystem. Could it be a pollinator attractant emitted by flowers (Misztal et al., 2010) or an insect deterrent emitted by conifers (Bouvier-Brown et al., 2009)? An additional minor suggestion would be to place p-allylanisole and p-cymene in a different category because these compounds are not strictly monoterpenes. You could consider something like "biogenic benzenoid" or "monoterpene-related BVOCs". AMCH is not strictly a monoterpene either but can be considered an oxygenated terpene.

The occurrence of potential sources of estragole (p-allylanisole) in this region could be confirmed by our botanist co-authors on this paper. Foeniculum vulgare (fennel) is an abundant and common weed in this study region. In addition, pine trees are common foreign tree species that are planted on farms, while numerous palm trees occurs in cities/towns surrounding Welgegund. Furthermore, and also linking to Comment 6, p-allylanisole emissions could also potentially have floral origin. Floral emissions would typically occur from October to February in this region. It is well-known that meadows in South Africa/this region have a significant number of species that flowers. South African grasslands are considered to be among the most diverse in the world, since it is primary grasslands, i.e. not man-made. In addition, these floral emissions could also be a potential source for other monoterpenes, which also could explain the relatively low isoprene concentrations compared to the relatively abundant monoterpenes. The following paragraphs have been included in Section 3.4:

"…within a 1 to 2 km radius, as indicated in Figure 2. The high monoterpene concentrations determined during the second sampling campaign may be associated with specific monoterpene emitting plants in the region.

Floral emissions could also be considered a potential source of monoterpenes in this region, which could also contribute to the relatively abundance of monoterpenes compared to the relatively low isoprene concentrations. Floral emissions in this region would typically occur with the onset of the wet season in October up until February. It is well-known that meadows, i.e. grazed grasslands in South Africa/this region have a significant number of species that flower. South African grasslands are considered to be exceptionally species rich (Siebert, 2011), since it is ancient, primary grasslands, i.e. not man-made (Bond, 2016).

Of particular interest is the potential sources of 4-allylanisole (estragole) due to its relatively substantial levels as indicated in Table 2. Bouvier-Brown et al. (2009a) and Misztal, et al., (2010) indicated that this species could potentially have a significant contribution to regional atmospheric chemistry due to relatively large estragole emissions measured from ponderosa pine trees and oil palms, respectively. As mentioned previously, pine trees are typically found on farms in this region as intruder tree species (Rouget, 2002), while numerous palm trees occurs in cities/towns surrounding Welgegund (Lubbe *et al.*, 2011). In addition, *Foeniculum vulgare* (fennel) – considered a typical source of estragole – is an abundant and common weed in this study region (Lubbe *et al.*, 2010). Furthermore, estragole emissions could also potentially have a floral origin."

In addition, p-allylanisole, p-cymene and AMCH were classified as "monoterpene-related BVOCs" in Table 2. The text in Section 2.3.1 paragraph 3 was also changed as follows:

"…and 2-methyl-3-butene-2-ol (MBO) with MDL between 0.9 and 1.4 pptv. The monoterpenes (α-pinene, camphene, β-pinene, $\Delta^3$-carene, limonene, 1,8-cineol, terpinolene, nopinone and bornylacetate) and monoterpene-related BVOCs (p-cymene, 4-allylanisole and 4-acetyl-1-methylcyclohexene (AMCH)) MDL was between 0.6 and 1.6 pptv. The term "monoterpene(s)" used in the discussions in subsequent sections in the manuscript refers to both the monoterpene and monoterpene-related BVOCs."

5) Have you observed any monoterpenes (or other BVOCs) related to stress? For example _-ocimene, methyl salicylate, green leaf volatiles?

The compounds mentioned in this comment were not included in our standard, but notable amounts were not detected.

6) Could some emissions at the Welgegund site have floral origin which could further explain why isoprene is relatively low whereas monoterpenes (and potentially other compounds such as p-allylanisole) are relatively abundant? Is the flowering happening an entire year round or seasonally? It would be very interesting because floral BVOCs from meadow-like flowering ecosystems can sometimes be abundant but receive relatively little attention compared to foliar emissions.

This comment was addressed in our response to Comment 4.

7) Table 4 contains interesting correlations, in particular, that MBO correlates with monoterpenes. Are these compounds coming from a conifer-like sources? On the other hand, I wonder if the result of isoprene correlating with MBO is more unexpected and it is also not discussed. Baker et al. 2001 found that if MBO is thermally treated (as is the case in GC) it can dehydrate and be detected as isoprene. Do you think this could be the case? While this is not meant to be a criticism, and given the different wind-roses and dependence on soil moisture/temperature perhaps the issue was probably minor but I still think it is worth giving this potential issue a general thought and discuss implications for isoprene/MBO data interpretation.

As discussed in Section 3.4, wind direction did not indicate any significant differences in the concentrations of MBO, monoterpenes and SQTs. As indicated in the discussion, it is difficult to point to specific sources of these species. However, in our response to Comment 4, it is mentioned that pine trees are common foreign tree species that are planted on farms. Therefore, pine trees could be considered a possible conifer-like source of MBO and monoterpenes. The following sentence was added in Section 3.4 (paragraph 2):

"…most likely to be the main sources, which can possibly include the urban footprint indicated in this region. In addition, pine trees are common foreign tree species that are planted on farms in this region (Rouget, 2002), which could be potential sources of MBO and monoterpenes."

As discussed in Section 3.2 (paragraph 4): "Guenther (2013) indicated that MBO is emitted from most isoprene emitting vegetation at an emission rate of ∼1 % of that of isoprene. However, MBO measured at Welgegund was approximately 30 % of the isoprene concentrations, which indicated that the main source of MBO at Welgegund is not from isoprene emitters, but from other MBO emitters." Furthermore, the possible dehydration of MBO to isoprene was considered to be insignificant, since when known amounts of MBO are injected into adsorbent tubes and analyzed with the analytical setup used in this study, <3% of isoprene is detected compared to MBO. Therefore as Referee#1 indicates, this correlation was unexpected and not further discussed.

8) Figure 4. This is an interesting figure. It seems that it is adopted from a different manuscript but clearly shows beautiful CO2 assimilation during the day and respired carbon during the night. If the data were available, it might be worth coloring these markers by PAR to better visualize assimilation vs respiration vs potentially anthropogenic CO2(?).

We thank Referee #1 for suggesting this improved visualisation of the figure. The markers in this figure were coloured with PPFD to better visualise assimilation vs respiration. Fig. 4 was replaced with the figure below:

[Figure]

Figure 4. Micrometeorological $CO_2$ flux measurements at Welgegund (Räsänen et al., 2016). The colour bar indicates the Photosynthetic Photon Flux Density (PPFD).

The following sentence was added in the text in Section the 3.1:

"…uptake of $CO_2$ by vegetation. In addition, the Photosynthetic Photon Flux Density (PPFD) is also indiacted with a colour bar. Negative values (downward $CO_2$ flux) indicate…"

9) It is mentioned several times (abstract, P22 L15 and in other places) that isoprene concentrations were higher from the western direction. "western direction" is not very informative for a reader in particular in the abstract. Careful reading points to the sentence in P5 L11 that ". . .western sector contains no major point sources and can therefore be considered to be representative of a relatively clean regional background". Maybe you meant to say that this direction does not contain any *anthropogenic* point sources of isoprene? Otherwise I wonder where this isoprene is coming from? If isoprene concentrations exhibit temperature dependence, it implies biogenic source but if there is no vegetation to the west, could there be a different source (e.g. heated rubber?). My suggestion would be simply to expand more clearly on the potential sources of the western isoprene.

The term "major point sources" in Section 2.1 does refer to "major anthropogenic point sources" and we thank Referee #1 for highlighting out this small, but significant technical point that creates confusion. Although not impossible, it is unlikely that isoprene measured at Welgegund will have an anthropogenic source e.g. heated rubber. The text was changed in Section 2.1 as follows:

"…Johannesburg-Pretoria conurbation (Tiitta, et al., 2014). From Figure 1, it is also evident that the western sector contains no major anthropogenic point sources and can therefore be considered to be representative of a relatively clean regional background."

With regard to expanding more clearly on the potential sources of the western isoprene, we consider that this topic is adequately explained in the manuscript in paragraphs 2 and 3 in Section 3.3. As argued in these paragraphs, it seems that the isoprene is mainly from the savanna grassland regions. This is indicated by the differences in the seasonal patterns of isoprene during the two sampling campaigns. A relatively distinct seasonal pattern was observed during the first sampling campaign for which the concentration rose indicated higher isoprene concentrations associated with winds originating in the south-western to southern sector from the grassland region in close proximity of Welgegund (Figure 2). During the second sampling campaign isoprene was only associated with the south-western to northern sector, i.e. a region with cultivated land in close proximity of Welgegund (Figure 2). The correlation between higher wind speed and isoprene was therefore considered to reflect isoprene emissions from a more distant savanna grassland fetch region in the south-western to northern sector.

We also agree with Referee #1 that the term "western direction" is not very informative for the reader in the context of the abstract, as well as in Section 3.4 (paragraph 2) and in the "Conclusions" section. Therefore this term was replaced…

in the Abstract as follows:

"…concentrations measured at Welgegund. An analysis of concentrations by wind direction indicated that isoprene concentrations were higher from the western sector that is considered to be a relatively clean regional background region with no large anthropogenic point sources (Figure 1), while wind direction did not indicate any significant differences in the concentrations of the other BVOC species. Statistical analysis indicated that soil moisture…",

and in Section 3.4 as follows:

"…concentrations, roses were compiled, as presented in Figure 6. In general, the concentration roses indicated that isoprene concentrations were higher from the western sector (indicated by the average and highest concentrations) that is considered to be a relatively clean regional background region with no large anthropogenic point sources, while wind direction did not indicate any significant differences in the concentrations of the other BVOC species. On occasion, higher MBO, monoterpene and SQT concentrations…"

The "Conclusions" section was re-written as suggested by Referee #1 in Comment 12 and the term "western direction" was not included in the revised "Conclusions" section.

10) I understand the median is often used to represent more episode-free concentration scenario. However, isoprene is only emitted during the day, so does it still make sense to show the monthly median for isoprene? Because you were collecting data both during the day and at night, I think it could be very interesting to separate day and night concentrations. In particular because monoterpenes unlike isoprene can often accumulate during the night in a shallow boundary layer so the overall median (or mean) concentration differences between the compounds (e.g. isoprene and monoterpenes) may not reflect strictly their emission strengths or true variability. I wonder if looking at some of the episodic events of high concentrations would not be an even more interesting opportunity to understand the chemistry scenarios.

This comment was addressed as part of our response to Comment 2 where it was indicated that median (mean) daytime to night-time concentrations ratios of BVOCs were included in Table 2 in order to separate daytime and night-time concentrations. These daytime to night-time concentrations ratios revealed the expected light dependency of isoprene for isoprene levels measured during the first sampling campaign, while no significant differences were observed for the other BVOCs measured (except for two monoterpenes and one SQT). Therefore, only the isoprene seasonal pattern plotted in the left panel in Figure 5(a) was replaced with seasonal patterns measured during daytime and night-time. The following sentences was included in the first paragraph in Section 3.3:

"…wet (October to April) and dry (May to September) season concentrations of the respective compounds measured for the two campaigns. As indicated in Section 3.2, isoprene measured during

the first sampling campaign had higher median (mean) daytime concentrations compared to median (mean) night-time concentrations, which reflects the light dependency expected from isoprene. All other BVOCs with the exception of two monoterpenes and one SQT did not indicate significant differences between daytime and night-time median (mean) concentrations. Therefore the seasonal plots of only isoprene were separated between daytime and night-time median concentrations."

11) Further to the point above, you are talking about ozone formation potentials from these BVOCs but maybe it would be worthwile to show some oxidant data. I am just curious how ozone (and also NOx, SO2, etc.) concentrations varied during the daytime and nighttime VOC sampling times and if it could teach us anything about the chemistry at the site.

As indicated in our response to Comment 1, a paper is currently being prepared where the reactions of all the VOCs (anthropogenic and biogenic) with oxidants, e.g. OH, $O_3$ will be further explored with atmospheric models in order to establish, for instance, whether $O_3$ formation within the region is VOC- or $NO_2$-limited. This paper will also incorporate the diurnal patterns of $O_3$ and $NO_2$. This work is included as future perspectives in the "Conclusions" section as indicated in our response at Comment 1.

12) It is recommended that the conclusions are made more succinct and emphasize major take-home messages which should be even more impressive than just the summary of the paper. For example, one could consider concluding about the implications for atmospheric chemistry and air quality in the region. In particular, the last two sentences of conclusions are unclear but the synopsis of future measurements is definitely needed to attract more attention and support more measurements in these almost completely unexplored regions of Africa.

We thank Referee #1 for suggesting an improved emphasised take-home message in this section. Therefore the Conclusion section was rewritten in order to be more succinct and to better emphasise the major take-home messages as follows:

"The annual median concentrations of isoprene, MBO, monoterpene and SQT determined during two sampling campaigns indicated that the sum of the concentrations of the monoterpenes was an order of magnitude higher than the concentrations of other BVOC species, with α-pinene being the most abundant species. Although monoterpene concentrations were similar to levels measured at other regions in the world and in a South Africa, very low isoprene concentrations at Welgegund led to a significantly lower total BVOC concentration compared to levels reported in most previous studies. In addition, total BVOC concentrations were an order of magnitude lower compared to the total aromatic VOC concentrations measured by Jaars et al. (2014) at Welgegund.

Distinct seasonal patterns were observed for MBO during both sampling campaigns, which coincided with wet and warmer months. Although less pronounced, a similar seasonal trend than for MBO was observed for isoprene during the first sampling campaign, while higher isoprene concentrations during the second sampling campaign were associated with higher wind speeds that indicated a distant source region of isoprene. No distinct seasonal pattern was observed for monoterpene and SQT concentrations. However, significantly higher levels of monoterpene and SQT, as well as MBO were measured from February to April 2011 during the first sampling campaign, which were attributed to the considerably higher soil moisture measured at a depth of 20 cm resulting for the wet season preceding the first campaign and is indicative of biogenic emissions from deep-rooted plants.

Woody species in the grassland region were considered to be the main sources of BVOCs measured, while sunflower and maize crops were also considered to be potential sources for BVOCs in this region. Multilinear regression analysis indicated that soil moisture had the most significant impact on atmospheric levels of MBO, monoterpene and SQT concentrations, while temperature had the greatest influence on isoprene levels.

The $O_3$ formation potentials of the BVOCs measured were an order of magnitude smaller than that determined for anthropogenic VOCs measured at Welgegund. Isoprene and the monoterpenes: ⏎ pinene, p-cymene, limonene and terpinolene, had the largest contribution to $O_3$ formation potential. ⏎ Pinene and limonene had the highest reaction rates with $O_3$, while isoprene exhibited relatively small contributions to the $O_3$ depletion. Limonene, pinene and terpinolene had the largest contributions to the OH-reactivity of BVOCs."

In addition, the last two sentences (last paragraph) were also improved as follows:

"It is important in future work that a comprehensive study on BVOC emissions from specific plant species in the area surrounding Welgegund must be performed in order to relate the emission capacities of vegetation types to the atmospheric BVOCs measured. It is also recommended that the oxidation products of BVOC species are measured in order to verify distant source regions of BVOCs measured at Welgegund."

and as indicated in our response to Comment 1, the following was also added to the synopsis of future work:

"In addition, the interactions between anthropogenic and biogenic VOCs must also be further explored, together with other ancillary measurements conducted at Welgegund (e.g. $SO_2$, $NO_2$ and $O_3$). Future work must also include investigating the reactions of all the VOCs measured with atmospheric oxidants (e.g. $^{\bullet}OH$ and $O_3$) with atmospheric chemistry models in order to establish, for instance, whether $O_3$ formation within the region is VOC- or $NO_2$-limited."

References:

Baker, B., Guenther, A., Greenberg, J., and Fall, R.: Canopy Level Fluxes of 2-Methyl-3-buten-2-ol, Acetone, and Methanol by a Portable Relaxed Eddy Accumulation System, Environ Sci Technol, 35, 1701-1708, 10.1021/es001007j, 2001.

This reference was not added in the manuscript as indicated in our response to Comment 7.

Bouvier-Brown, N. C., Goldstein, A. H., Worton, D. R., Matross, D. M., Gilman, J. B., Kuster, W. C., Welsh-Bon, D., Warneke, C., de Gouw, J. A., Cahill, T. M., and Holzinger, R.: Methyl chavicol: characterization of its biogenic emission rate, abundance, and oxidation products in the atmosphere, Atmos. Chem. Phys., 9, 2061-2074, 2009.

This reference was added in the manuscript as indicated in our response to Comment 4.

Misztal, P. K., Owen, S. M., Guenther, A. B., Rasmussen, R., Geron, C., Harley, P., Phillips, G. J., Ryan, A., Edwards, D. P., Hewitt, C. N., Nemitz, E., Siong, J., Heal, M. R., and Cape, J. N.: Large estragole fluxes from oil palms in Borneo, Atmos. Chem. Phys., 10, 4343-4358, 10.5194/acp-10-4343-2010, 2010.

This reference was added in the manuscript as indicated in our response to Comment 4.

---

## Author Comment (AC2) · 7 Nov 2016

Summary: The authors present two separate years of atmospheric measurements of biogenic volatile organic compounds (BVOCs) at a grazed savannah-grasslandagriculture landscape in South Africa. This long-term and chemically detailed data set allows the authors to accurately assess the seasonal variability of these reactive compounds and provides the scientific community with a valuable dataset of BVOC emissions in a rarely studied ecosystem. The author's present a careful comparison of the observed emissions to the surrounding vegetation and other physical parameters such as soil moisture. The contributions of each species to potential ozone formation are explored. The data is of high quality and the manuscript is very well written. I suggest publication with only minor revisions.

We would like to thank Referee #2 for the positive review of this paper, as well as acknowledging the relevance of the work presented and the high quality of the dataset. We would also like to thank Referee #2 for the minor technical comments, which were each addressed. Below is a point-by-point response to each of these comments.

Technical comments (P=page number and L=line number):

P1L22 and P3L5: These are not exactly the same statements. The first infers that anthropogenic sources only contribute 10% of the global annual VOC budget. The second statement (referencing Guenther et al.) states that 90% of BVOC emissions are from vegetation/terrestrial sources (i.e., only 10% from oceanic sources). Please be sure that these distinctions are more clear in the manuscript to avoid confusion by clarifying the contribution of BVOC vs. anthropogenic VOC. Also, do the emissions refer to mass or carbon or some other unit?

We thank Referee #2 for pointing out the confusion in these statements. It was meant to be stated that 90% of BVOC emissions are from vegetation. In addition, the emissions refer to mass per annum. Therefore the sentences in the Abstract and Introduction was changed as follows:

"…one ($O_3$) and secondary organic aerosols (SOA). Ecosystems produce and emit a large number of BVOCs. It is estimated on a global scale that approximately 90 % of annual BVOC emissions are from terrestrial sources. In this study, measurements…"

"…with typically 0.2 to 10 % of the carbon uptake during photosynthesis being converted to BVOCs (Kesselmeier et al., 2002). It is estimated that, on a global scale, approximately 90 % of annual BVOC emissions are from vegetation/ terrestrial sources (~1000 Tg year$^{-1}$) (Guenther et al., 2012)."

P2L12: lower for the grassland savannah or the other landscapes?

It was lower for grassland savannah. The sentence was changed as follows in the Abstract:

"However, comparisons with measurements conducted at other landscapes in southern Africa and the rest of the world that have more woody vegetation indicated that BVOC concentrations were, in general, significantly lower for the grassland savannah."

P4L25 and P29L2: African to Africa

The correction was made as follows:

"…concentrations covering a full seasonal cycle in southern Africa and for a grassland bioregion…"

P8L3: Can you state what the efficiency of the ozone removal was?

The efficiency of the ozone removal was stated in the paper by Jaars et al., 2014. However, we agree that it will be good to also indicate it in this paper. The following was added in Section 2.3.1:

"At regular intervals, the efficiency of this $O_3$ removal was verified with an $O_3$ monitor, which indicated that $O_3$ concentrations decreased from median values $\geq$ 30 ppb to < 2 ppb (Jaars et al., 2014)."

Section 2.3.1. Please add a brief description on how the samples were stored/transported prior to analysis. How much time would elapse between collection and analysis?

Handling of the samples after sampling and prior to analysis were also discussed in detail in the paper by Jaars et al., 2014. We also agree with Referee #2 that a brief summary can be included in this paper and the following was added in Section 2.3.1:

"…helium for 30 minutes at 350 ℃ at a flow of 40 ml min$^{-1}$. The adsorbent tubes were removed from the sampler once a week and closed off with Swagelok® caps. Each tube was stored in a container for transport to the laboratory, where the adsorbent tubes were stored in a freezer for two to four weeks prior to analysis."

Section 3.5. How much of the correlation is simply driven by the fact that MBO, MTs, and SQTs generally have higher concentrations at night while isoprene will have the largest daytime emission? It

would be interesting to compare the day and night time values of the compound classes, perhaps just for the wet season when emissions were enhanced.

In our response to a comment of Referee #1, median (mean) daytime to night-time concentrations ratios of all the BVOCs measured during both sampling campaigns were included in Table 2. These daytime to night-time concentration ratios indicated that there were no significant differences in levels of most of the BVOCs measured during daytime and night-time at Welgegund, with the exception of isoprene measured during the first sampling campaign, as well as two monoterpenes (terpinolene, bornylacetate) and the SQT aromadendrene measured duirng the second sampling campaign. Therefore isoprene levels measured during the first sampling campaign reflected the light dependency usually associated with isoprene emissions. However, daytime to night-time concentration ratios of isoprene did not exhibited the very strong light dependency typically associated with atmospheric isoprene concentrations, which could be attributed to the characteristics of sources of these species. Therefore the light dependency associated with these few BVOCs measured at Welgegund was addressed in the first paragraph in Section 3.2 as follows:

"In Table 2, the median (mean) and inter-quartile range (IQR, $25^{th}$ to $75^{th}$) concentrations, as well as the median (mean) daytime to night-time concentration ratios of the BVOC species determined during the two sampling campaigns at Welgegund are presented. It is evident from the median (mean) daytime to night-time concentration ratios that there were not significant differences in levels of most of the BVOCs measured during daytime and night-time at Welgegund, with the exception of isoprene measured during the first sampling campaign, as well as the monoterpenes terpinolene and bornylacetate, and the SQT aromadendrene measured duirng the second sampling campaign. Isoprene levels during the first sampling campaign were approximately two times higher during daytime, which reflect the light dependency usually associated with isoprene emissions. However, daytime to night-time concentration ratios of isoprene did not exhibited the strong light dependency typically associated with atmospheric isoprene concentrations, which could be attributed to the characteristics of sources of these species that are discussed in subsequent sections. The temperature and photoactive radiation (PAR) measurements at Welgegund was used in the MEGAN BVOC emission model, which indicated that the measurement time (11:00 to 13:00 local time) captured most of the period of maximum isoprene emission (typically about 12:00 to 2:00 local time). In addition, by assuming a typical diurnal variation in VOC oxidation rate and boundary layer height, it was also found that the isoprene concentration of the measurement time is representative of the daytime isoprene concentration (Greenberg et al., 1999). In Table 3, the concentrations of BVOC species measured during other campaigns in South Africa and the rest of the world are presented."

Furthermore, based on these daytime to night-time concentration rations, it also does not seem that the observed correlations are driven by daytime and night-time emissions.